# ANTITHETIC NOISE IN DIFFUSION MODELS

**Jing Jia**[1]   **Sifan Liu**[2*]   **Bowen Song**[3]   **Wei Yuan**[1]   **Liyue Shen**[3*]   **Guanyang Wang**[1*]

[1]Rutgers University, `{jing.jia, wy204, guanyang.wang}@rutgers.edu`
[2]Duke University, `sifan.liu@duke.edu`
[3]University of Michigan, `{bowenbw, liyues}@umich.edu`

## ABSTRACT

We systematically study antithetic initial noise in diffusion models, discovering that pairing each noise sample with its negation consistently produces strong negative correlation. This universal phenomenon holds across datasets, model architectures, conditional and unconditional sampling, and even other generative models such as VAEs and Normalizing Flows. To explain it, we combine experiments and theory and propose a *symmetry conjecture* that the learned score function is approximately affine antisymmetric (odd symmetry up to a constant shift), supported by empirical evidence. This negative correlation leads to substantially more reliable uncertainty quantification with up to $90\%$ narrower confidence intervals. We demonstrate these gains on tasks including estimating pixel-wise statistics and evaluating diffusion inverse solvers. We also provide extensions with randomized quasi-Monte Carlo noise designs for uncertainty quantification, and explore additional applications of the antithetic noise design to improve image editing and generation diversity. Our framework is training-free, model-agnostic, and adds no runtime overhead. Code is available at `https://github.com/jjia131/Antithetic-Noise-in-Diffusion-Models-page`.

## 1 INTRODUCTION

Diffusion models have set the state of the art in photorealistic image synthesis, high-fidelity audio, and video generation (Sohl-Dickstein et al., 2015; Ho et al., 2020; Song et al., 2021b; Kong et al., 2021); they also power applications such as text-to-image generation (Rombach et al., 2022), image editing and restoration (Meng et al., 2022), and inverse problem solving (Song et al., 2022).

For many pretrained diffusion models, sampling relies on three elements: the network weights, the denoising schedule, and the initial Gaussian noise. Once these are fixed, sampling is often deterministic: the sampler transforms the initial noise into an image via successive denoising passes.

Much of the literature improves the first two ingredients and clusters into two strands: (i) architectural and training developments, which improve sample quality and scalability through backbone or objective redesign (e.g., EDM (Karras et al., 2022), Latent Diffusion Models (Rombach et al., 2022), DiT (Peebles & Xie, 2023)), (ii) accelerated sampling, which reduces the number of denoising steps while retaining high-quality generation (e.g., DDIM (Song et al., 2021a), Consistency Models (Song et al., 2023), DPM-Solver++ (Lu et al., 2023), and progressive distillation (Salimans & Ho, 2022)).

However, the third ingredient—the *initial Gaussian noise*—has received comparatively little attention. Prior work has optimized initial noise for generation quality, editing, controllability, or inverse problem solving (Guo et al., 2024; Qi et al., 2024; Zhou et al., 2024; Ban et al., 2025; Chen et al., 2024a; Eyring et al., 2024; Song et al., 2025; Wang et al., 2024; Chihaoui et al., 2024). However, most of these efforts are task-specific. A systematic understanding of how noise itself shapes diffusion model outputs is still missing.

Our perspective is orthogonal to prior work. Our central discovery is both *simple and universal*: pairing every Gaussian noise $z$ with its negation $-z$—known as *antithetic sampling* (Owen, 2013)—consistently produces samples that are **strongly negatively correlated**. This phenomenon holds

---
*co-corresponding authors

regardless of architecture, dataset, sampling schedule, and both conditional and unconditional sampling. It further extends to other generative models such as VAEs and Normalizing Flows. We explain this phenomenon through both experiments and theory. This leads to a symmetry conjecture that the score function is approximately affine antisymmetric, providing a new structural insight supported by empirical evidence.

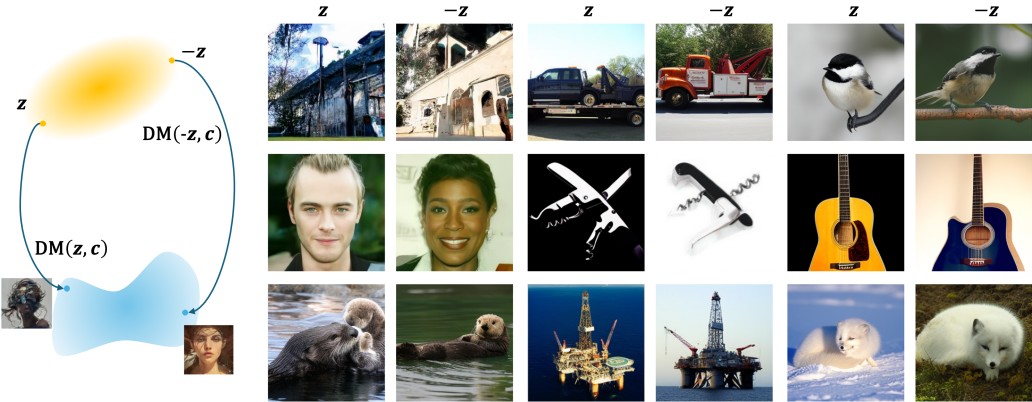

Figure 1: Use antithetic noise $-z$ and $z$ (with condition $c$) to generate visually "opposite" images.

This universal property has direct impact on uncertainty quantification and also enables additional applications:

**(i) Sharper uncertainty quantification.** Antithetic pairs naturally act as control variates, enabling significant variance reduction and thus sharper uncertainty quantification. Our antithetic estimator delivers up to $90\%$ tighter confidence intervals and cuts computation cost by more than 100 times. The efficiency gain immediately leads to huge cost savings in a variety of tasks, including bias detection in generation and diffusion inverse solver evaluation, as we demonstrate in later experiments.

**(ii) Other applications.** Because each antithetic noise pair drives reverse-diffusion trajectories toward distant regions of the data manifold (see Figure 1), the paired-sampling scheme increases diversity "for free" while preserving high image quality, as confirmed by SSIM and LPIPS in our experiments. Moreover, algorithms that rely on intermediate sampling or approximation steps also benefit from the improved reliability provided by antithetic noise. As an illustration, we present an image editing example in C.2 and show that our antithetic design serves as a *plug-and-play* tool to improve performance at no additional cost.

Building on the antithetic pairs, we generalize the idea to apply quasi-Monte Carlo (QMC) and randomized QMC (RQMC). The resulting RQMC estimator often delivers further variance reduction. Although QMC has been widely used in computer graphics (Keller, 1995; Waechter & Keller, 2011), quantitative finance (Joy et al., 1996; L'Ecuyer, 2009), and Bayesian inference (Buchholz et al., 2018; Liu & Owen, 2021), this is, to our knowledge, its first application to diffusion models.

In summary, we discover a universal property of initial noise, reveal a new symmetry in score networks, and demonstrate concrete benefits in various practical applications. This positions initial noise manipulations as a simple, training-free tool for advancing generative modeling.

The remainder of the paper is organized as follows. Section 2 defines the problem and outlines our motivation. Section 3 presents our central finding that antithetic noise pairs produce strongly negatively correlated outputs. We offer both theoretical and empirical explanations for this phenomenon, and present the *symmetry conjecture* in Section 3.2. Section 4 develops estimators and their confidence intervals via antithetic sampling, and extends the approach to QMC. Section 5 reports experiments on the aforementioned applications. Section 6 discusses our method and outlines future directions. Appendix B, C, and D include proofs, additional experiments and detailed setups, and supplemental visualizations, respectively.

## 2 SETUP, MOTIVATION, AND RELATED WORKS

**Unconditional diffusion model:** A diffusion model aims to generate samples from an unknown data distribution $p_0$. It first *noises* data towards a standard Gaussian progressively, then learns to reverse the process so that Gaussian noise can be *denoised* step-by-step back into target samples. The forward process simulates a stochastic differential equation (SDE): $d\mathbf{x}_t = \mu(\mathbf{x}_t, t)dt + \sigma_t d\mathbf{w}_t$, where $\{\mathbf{w}_t\}_{t=0}^T$ denotes the standard Brownian motion, and $\mu(\mathbf{x}, t), \sigma_t$ are chosen by the users. Let $p_t$ denote the distribution of $\mathbf{x}_t$. Song et al. (2021b) states that if we sample $\mathbf{y}_T \sim p_T$ and simulate the probability-flow ordinary differential equation (PF-ODE) backward from time $T$ to 0 as

$$d\mathbf{y}_t = \left( -\mu(\mathbf{y}_t, t) - \frac{1}{2}\sigma_t^2 \nabla \log p(\mathbf{y}_t, t) \right) dt, \tag{1}$$

then for every time $t$ the marginal distribution of $\mathbf{y}_t$ coincides with $p_t$. Thus, in an idealized world, one can perfectly sample from $p_0$ by simulating the PF-ODE (1).

In practice, the score function $\nabla \log p_t(\mathbf{x}, t)$ is unavailable, and a neural network $\epsilon_\theta^{(t)}(\mathbf{x})$ is trained to approximate it, where $\theta$ denotes its weights. Therefore, one can generate new samples from $p_0$ by first sampling a Gaussian noise and simulating the PF-ODE (1) through a numerical integrator from $T$ to 0. For example, DDIM (Song et al., 2021a) has the (discretized) forward process as $\mathbf{x}_k \mid \mathbf{x}_{k-1} \sim \mathbb{N}(\sqrt{\alpha_k}\mathbf{x}_{k-1}, (1-\alpha_k)I)$ for $k = 0, \ldots, T-1$, and backward sampling process as

$$\mathbf{y}_T \sim \mathbb{N}(0, I), \quad \mathbf{y}_{t-1} = \sqrt{\alpha_{t-1}}\left( \frac{\mathbf{y}_t - \sqrt{1-\alpha_t}\,\epsilon_\theta^{(t)}(\mathbf{y}_t)}{\sqrt{\alpha_t}} \right) + \sqrt{1-\alpha_{t-1}}\,\epsilon_\theta^{(t)}(\mathbf{y}_t). \tag{2}$$

Once $\theta$ is fixed, the randomness in DDIM sampling comes solely from the initial Gaussian noise.

We remark that samples from $p_0$ can also be drawn by simulating the backward SDE with randomness at each step, as in the DDPM sampler (Ho et al., 2020; Song et al., 2021b). Throughout the main text, we focus on deterministic samplers such as DDIM to explain our idea. In Appendix C.4, we present additional experiments showing that our findings also extend to stochastic samplers like DDPM.

**Text–conditioned latent diffusion:** In Stable Diffusion and its successors SDXL and SDXL Turbo (Podell et al., 2024; Sauer et al., 2024), a pretrained VAE first compresses each image to a latent tensor $z$. A text encoder embeds the prompt as $c \in \mathbb{R}^m$. During training, the network $\epsilon_\theta^{(t)}$ receives $(z_t, c)$ and learns $\nabla_{z_t} \log p_t(z_t \mid c)$. At generation time we draw latent noise $z_T \sim \mathbb{N}(0, I)$ and run the reverse diffusion from $t = T$ to 0, yielding a denoised latent $z_0$, which the decoder maps back to pixels. Given a prompt $c$ and an initial Gaussian noise, the sampler produces an image from $p_0(\cdot \mid c)$.

**Diffusion posterior sampling:** Diffusion model can also be used as a prior in inverse problems. Suppose we observe only a partial or corrupted measurement $\mathbf{y}_{\text{obs}} = \mathcal{A}(\mathbf{x}) + \text{noise}$ from an unknown signal $\mathbf{x}$. Diffusion posterior sampling aims to sample from the posterior distribution $p(\mathbf{x} \mid \mathbf{y}_{\text{obs}}) \propto p(\mathbf{y}_{\text{obs}} \mid \mathbf{x}) p(\mathbf{x})$, where $p(\mathbf{x})$ is given by a pretrained diffusion model. Given initial noise $z$ and observed $\mathbf{y}_{\text{obs}}$, diffusion posterior samplers apply a sequence of denoising steps from $T$ to 0, each step resembling (2) but incorporating $\mathbf{y}_{\text{obs}}$ (Chung et al., 2023a; Song et al., 2022; 2024).

### 2.1 MOTIVATION

Beyond examining how variations in the initial noise affect the outputs, several technical considerations motivate our focus on the antithetic noise pair $(z, -z)$ for diffusion models. Let DM denote certain diffusion model's mapping from an initial noise vector to a generated image sample.

- **Preserving quality:** Since $z \sim \mathbb{N}(0, I)$ implies $-z \sim \mathbb{N}(0, I)$, the initial noises $z$ and $-z$ share the same marginal distribution. Consequently, $\text{DM}(z) \overset{d}{=} \text{DM}(-z)$, so all per-sample statistics remain unchanged. In other words, negating the initial noise does not degrade generation quality.

- **Maximal separation in the noise space:** High-dimensional Gaussians concentrate on the sphere of radius $\sqrt{\text{dimension}}$ (Vershynin, 2018), so any draw $z$ and its negation $-z$ lie at opposite poles of that hypersphere. This antipodal pairing represents the maximum possible perturbation in noise space, making it a natural extreme test of the sampler's behavior.

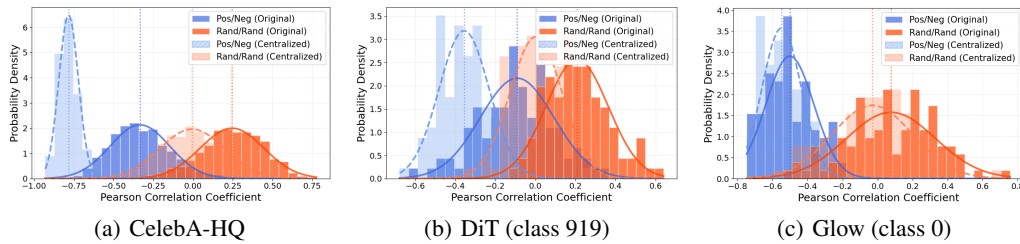

Figure 2: Histograms of standard and centralized Pearson correlation coefficients for CelebA-HQ, and DiT class and Glow in single class. Dashed lines indicate the average.

- **Measuring (non-)linearity:** Recent work has examined how closely score networks approximate linear operators. Empirical studies across various diffusion settings show that these networks behave as locally linear maps; this behavior forms the basis for new controllable sampling schemes (Chen et al., 2024b; Li et al., 2024b; Song et al., 2025). Correlation is a natural choice to measure linearity: if a score network were exactly linear, feeding it with noises $z$ and $-z$ would yield a correlation of $-1$ between $\mathrm{DM}(z)$ and $\mathrm{DM}(-z)$. Thus, the difference between the observed correlation and $-1$ provides a direct measure of the network's departure from linearity.

## 3 NEGATIVE CORRELATION FROM ANTITHETIC NOISE

### 3.1 PIXEL-WISE CORRELATIONS: ANTITHETIC VS. INDEPENDENT NOISE

We compare the similarity between paired images generated under two sampling schemes: PN (positive vs. negative, $z$ vs. $-z$) and RR (random vs. random, $z_1$ vs. $z_2$) under three settings: (1) *unconditional diffusion models*, (2) *class- or prompt-based conditional diffusion models*, and (3) *generative models beyond diffusion*.

For diffusion models, we evaluate both unconditional and conditional generation using publicly available pre-trained checkpoints. Notably, we include both traditional U-Net architecture and transformer-based DiT (Peebles & Xie, 2023). Implementation details are described in Section 5.1. We also test on distilled diffusion models (Song et al., 2023) using consistency distillation checkpoints. For generative models beyond diffusion, we select two representative baselines: an unconditional VAE on MNIST and a conditional Glow model (Kingma & Dhariwal, 2018) on CIFAR-10. The experimental details for consistency models, VAE and Glow are provided in Appendix C.1.

To quantify similarity, we use two metrics: the standard Pearson correlation and a centralized Pearson correlation. Let $x_{i,1}$ and $x_{i,2}$ denote the flattened pixel values of the two images in the $i$-th generated pair. The standard Pearson correlation is computed directly between $x_{i,1}$ and $x_{i,2}$. To correct for dataset-level or class-level bias, we also define a centralized correlation. For a dataset, prompt, or class with $K$ pairs generated, we compute the mean $\mu_c = \sum_{i=1}^{K}(x_{i,1} + x_{i,2})/2K$. The centralized correlation of the $i$-th pair is defined as the standard Pearson correlation between the centralized images $x_{i,1} - \mu_c$ and $x_{i,2} - \mu_c$. For each comparison, $t$-test is conducted to assess statistical significance. In all experiments, the resulting $p$-values are negligible ($< 10^{-10}$), which confirms significance; hence, they are omitted from the presentation.

Additional metrics such as the Wasserstein distance are presented in Appendix C.4.

**Results:** Table 1 summarizes the statistics in all classes of models. PN pairs consistently show significantly stronger negative correlations than RR pairs, with this contrast also visually evident in their histograms (Figure 2). In addition, centralization strengthens the negative correlation, since it removes shared patterns. For example, in CelebA-HQ, centralization removes global facial structure, while in DiT and Glow, it removes class-specific patterns.

The same behavior appears in both DDPM models and diffusion posterior samplers, which we report in Appendix C.4. These results demonstrate that the negative correlation resulting from antithetic sampling is a universal phenomenon across diverse architectures and conditioning schemes.

Table 1: Correlation results across different models and datasets, shown are means (SD). Rows 1–3 are pretrained unconditional diffusion models on different datasets. Rows 4–5 are conditional diffusion models. Rows 6–8 are pretrained consistency models on different datasets. Rows 9–10 are generative models that are not diffusion-based.

| Model | Dataset | Standard Correlation | | Centralized Correlation | |
|---|---|---|---|---|---|
| | | PN | RR | PN | RR |
| Uncon. Model | LSUN-Church | -0.62 (0.11) | 0.08 (0.17) | -0.77 (0.07) | 0.00 (0.17) |
| | CelebA-HQ | -0.34 (0.19) | 0.25 (0.20) | -0.78 (0.06) | -0.01 (0.20) |
| | CIFAR-10 | -0.76 (0.13) | 0.05 (0.24) | -0.86 (0.07) | 0.00 (0.23) |
| SD 1.5 | LAION-2B(en) | -0.47 (0.05) | 0.05 (0.04) | -0.54 (0.08) | 0.00 (0.01) |
| DiT | ImageNet-256 | -0.07 (0.27) | 0.26 (0.18) | -0.45 (0.08) | 0.00 (0.03) |
| Consistency Model | LSUN-Cat | -0.88 (0.06) | 0.02 (0.14) | -0.91 (0.05) | 0.01 (0.14) |
| | LSUN-Bedroom | -0.78 (0.07) | 0.03 (0.15) | -0.84 (0.06) | 0.00 (0.15) |
| | ImageNet-64 | -0.71 (0.14) | 0.02 (0.19) | -0.75 (0.12) | 0.01 (0.19) |
| VAE | MNIST | 0.21 (0.12) | 0.42 (0.17) | -0.41 (0.12) | -0.00 (0.24) |
| Glow | CIFAR-10 | -0.52 (0.02) | 0.08 (0.05) | -0.57 (0.02) | -0.01 (0.02) |

## 3.2 Explanatory experiments: temporal correlations & symmetry conjecture

We aim to explain the strong negative correlation between $DM(Z)$ and $DM(-Z)$ for $Z \sim \mathbb{N}(0, I)$. Because DM performs iterative denoising through the score network $\epsilon_\theta^{(t)}$ (see (2)), we visualize how these correlations evolve over diffusion time-steps in Figure 3.

Throughout the transition from noises to samples, the PN correlation of $\epsilon_\theta^{(t)}$—indicated by the orange, blue, and green dashed lines in Figure 3—starts at $-1$, stays strongly negative, and only climbs slightly in the final steps. This nearly $-1$ correlation is remarkable, as it suggests that the score network $\epsilon_\theta$ learns an approximately affine antisymmetric function. To explain, we first state the following results which shows $-1$ correlation is equivalent to affine antisymmetric, with proof in Appendix B.1.

**Lemma 1.** *Let $Z \sim \mathbb{N}(0, I_d)$, suppose a map $f : \mathbb{R}^d \rightarrow \mathbb{R}$ satisfies $\mathrm{Corr}(f(Z), f(-Z)) = -1$, then $f$ must be affine antisymmetric at $(0, c)$ for some c, i.e., $f(\mathbf{x}) + f(-\mathbf{x}) = 2c$ for every $\mathbf{x}$.*

Lemma 1 and Figure 3 leads us to the following conjecture:

**Conjecture** For each time step $t$, the score network $\epsilon_\theta^{(t)}$ is approximately affine antisymmetric at $(\mathbf{0}, \mathbf{c}_t)$, i.e.,

$$\epsilon_\theta^{(t)}(\mathbf{x}) + \epsilon_\theta^{(t)}(-\mathbf{x}) \approx 2\mathbf{c}_t$$

for some fixed vector $\mathbf{c}_t$ that depends on $t$.

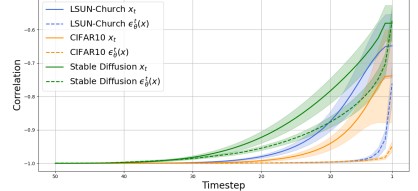

Figure 3: Correlation of $x_t$ (solid) and $\epsilon_\theta^{(t)}$ (dash) between antithetic (PN) pairs. Step 50 is the initial noise and Step 0 is the generated image. Shaded bands show $\pm 1$ std. dev.

The conjecture implies that the score network has learned an "almost odd" function up to an additive shift. This is consistent with both theory and observations for large $t$, where $p_t(\mathbf{x})$ is close to standard Gaussian, thus the score function is almost linear in $\mathbf{x}$. Nonetheless, our conjecture is far more general: it applies to every timestep $t$ and can accommodate genuinely nonlinear odd behaviors (for example, functions like $\sin x$).

This conjecture combined with the DDIM update rule (2) immediately explains the negative correlation: The one step iteration of $F_t$ in (2) can be written as a linear combination between $\mathbf{x}$ and the output of the score network: $F_t(\mathbf{x}) = a_t\mathbf{x} + b_t\epsilon_\theta^{(t)}(\mathbf{x})$. Given the conjecture, $F_t(-\mathbf{x}) \approx -a_t\mathbf{x} - b_t\epsilon_\theta^{(t)}(\mathbf{x}) + 2b_t\mathbf{c}_t = -F_t(\mathbf{x}) + 2b_t\mathbf{c}_t$, so the one-step DDIM update is affine antisymmetric at $(\mathbf{0}, b_t\mathbf{c}_t)$. Thus, beginning with a strongly negatively correlated pair, each DDIM update preserves that strong negative correlation all the way through to the final output.

To (partly) validate our conjecture, we perform the following experiment. Using a pretrained CIFAR-10 score network with a 50-step DDIM sampler, we picked time steps $t = 1, 3, \ldots, 19$ (where smaller $t$ is closer to the image and larger $t$ is closer to pure noise). For each $t$, we sample $\mathbf{x} \sim \mathbb{N}(0, I_d)$, evaluated the first coordinate of $\epsilon_\theta^{(t)}(c\,\mathbf{x})$ as $c$ varied from $-1$ to $1$ (interpolating from $-\mathbf{x}$ to $\mathbf{x}$), and plotted the result in Figure 4. We exclude $t \geq 20$, where the curve is very close to a straight line.

Figure 4 confirms that, at every step $t$, the first coordinate of the score network is indeed overall affine antisymmetric. Although the curves display nonlinear oscillations at small $t$, the deviations on either side are approximately mirror images, and for larger $t$ the mapping is almost a straight line. The symmetry center $c_t$ usually lies near zero, but not always—for instance, at $t = 11$, $c_t \approx 0.05$ even though the function spans from $-0.05$ to $0.15$. In Appendix C.5, we present further validation experiments using alternative coordinates, datasets, and a quantitative

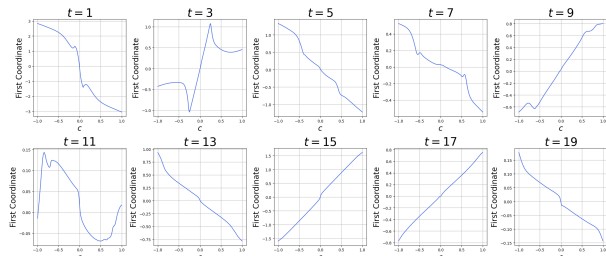

Figure 4: First-coordinate output of the pretrained score network on CIFAR10 as a function of the interpolation scalar $c$ for a 50-step DDIM at $t = 1, 3, \ldots, 19$.

metric called the antisymmetry score. Despite the distinct function shapes across coordinates, the conjectured symmetry still persists.

We further evaluate the antisymmetry conjecture across additional coordinates and datasets. On CIFAR-10 and Church, we assess how closely one-dimensional slices of the network outputs behave like affine-antisymmetric functions. To this end, we introduce the *affine antisymmetry score*, which quantifies the degree of antisymmetry. Across datasets, the resulting antisymmetry scores are consistently high (mean values above 0.99 and even the lower quantiles still near 0.9). These results provide strong empirical support for the conjecture; full experimental details and plots are given in Appendix C.5.

Finally, we provide theoretical support for our conjecture in Appendix B.2, B.3, and B.4. Appendix B.2 confirms the conjecture in the large-$t$ regime, Appendix B.3 shows how both the density ratio and the score error converge monotonically as $t$ evolves via a Hermite polynomial expansion, and Appendix B.4 demonstrates that all orthogonal symmetries of the data distribution are preserved throughout the forward process. These results help explain the behavior underlying the conjecture. The forward process preserves every orthogonal symmetry of the initial distribution, including coordinate reflections. Thus, in the ideal case where the data distribution at $t = 0$ is reflection-symmetric, the density remains even and the score remains odd for all $t$ by Proposition 4. Moreover, even if this symmetry holds only approximately for small $t$, the monotone decay of the density ratio and the score error shows that $p_t$ quickly approaches the Gaussian, so the forward dynamics push the score toward an odd function in a controlled manner.

Both the conjecture and its empirical support could be of independent interest. They imply that the score network learns a function with strong symmetry, even though that symmetry is not explicitly enforced. The conjecture matches the Gaussian linear score approximation in the high-noise regime, which has proven effective (Wang & Vastola, 2024). In the intermediate-to-low noise regime, a single Gaussian approximation performs poorly—this is clear in Figure 4 for $t = 3, 7, 11$, where the score is strongly nonlinear—yet our conjectured approximate symmetry still holds.

Besides explaining the source of negative correlation, we believe this result provides new insight into the structure of diffusion models, which are often treated as black-box functions. Leveraging this finding for algorithms and applications is an important direction for future work.

For readers interested in an additional theoretical perspective, Appendix B.6 presents a discussion that links negative correlation to the FKG inequality, and provides a detailed analysis of the DDIM sampler using this framework.

## 4 UNCERTAINTY QUANTIFICATION

In uncertainty quantification for a diffusion model DM, the goal is typically to estimate expectations of the form $\mathbb{E}_{z\sim\mathbb{N}(0,I)}\big[S\big(\mathrm{DM}(z)\big)\big]$, where $S$ is a statistic of user's interest. Our goal is to leverage negative correlation to design an estimator with higher accuracy at fixed computation cost of inference.

Recent studies have examined epistemic uncertainty and bias in diffusion models, including Berry et al. (2024; 2025). Our focus is different: we study aleatoric uncertainty from noise sampling and develop a variance-reduction method.

**Standard Monte Carlo:** To approximate this expectation, the simplest approach is to draw $N$ independent noises $z_1, \ldots, z_N \sim \mathbb{N}(0, I)$, calculate $S_i = S\big(\mathrm{DM}(z_i)\big)$ for each $i$ and form the standard Monte Carlo (MC) estimator $\hat{\mu}_N^{\mathrm{MC}} := \sum_{i=1}^{N} S_i/N$ by taking their average. A $(1 - \alpha)$ confidence interval, denoted as $\mathrm{CI}_N^{\mathrm{MC}}(1 - \alpha)$ is then $\hat{\mu}_N^{\mathrm{MC}} \pm z_{1-\alpha/2}\sqrt{(\hat{\sigma}_N^{\mathrm{MC}})^2/N}$, where $\hat{\sigma}_N^2$ is the sample variance of $S_1, S_2, \ldots, S_N$ and $z_{1-\alpha/2}$ is the $(1 - \alpha/2)$-quantile of the standard normal. A formal guarantee of the above construction is given in Proposition 5 in Appendix B.5.

**Antithetic Monte Carlo:** The observed negative correlation motivates an improved estimator. Let $N = 2K$ be even, users can generate $K$ pairs of antithetic noise $(z_1, -z_1), \ldots, (z_K, -z_K)$. Define $S_i^+ = S(\mathrm{DM}(z_i))$, $S_i^- = S(\mathrm{DM}(-z_i))$, and let $\bar{S}_i = 0.5(S_i^+ + S_i^-)$ be their average. Our Antithetic Monte Carlo (AMC) estimator is $\hat{\mu}_N^{\mathrm{AMC}} := \sum_{i=1}^{K} \bar{S}_i/K$, and confidence interval $\mathrm{CI}_N^{\mathrm{AMC}}(1 - \alpha)$ is $\hat{\mu}_N^{\mathrm{AMC}} \pm z_{1-\alpha/2}\sqrt{2(\hat{\sigma}_N^{\mathrm{AMC}})^2/N}$, where $(\hat{\sigma}_N^{\mathrm{AMC}})^2$ is the sample variance of $\bar{S}_1, \ldots, \bar{S}_K$.

The intuition is simple: if the pair $(S_i^+, S_i^-)$ remains negatively correlated, then averaging antithetic pairs can reduce variance by partially canceling out opposing errors—since negative correlation suggests that when one estimate exceeds the true value, the other is likely to fall below it.

The AMC estimator is unbiased, and $\mathrm{CI}_N^{\mathrm{AMC}}(1 - \alpha)$ achieves correct coverage as the sample size increases. Let $\rho := \mathrm{Corr}(S_i^+, S_i^-)$. We can prove that AMC's standard error and its confidence-interval width are each equal to the Monte Carlo counterparts multiplied by the factor $\sqrt{1 + \rho}$. Thus, when $\rho < 0$, AMC produces **provably lower variance and tighter confidence intervals** than MC, with greater gains as $\rho$ becomes more negative. See Appendix B.5 for a formal statement and proof.

Since both methods have the same computational cost, the variance reduction from negative correlation and the antithetic design yields a direct and cost-free improvement.

$K$**-antithetic noise:** We can generalize the antithetic noise pair to a collection of $K$ noise variables, constructed so that every pair has the same negative correlation $-1/(K-1)$. One way to generate them is as follows. Draw $K$ independent standard Gaussian vectors $(w_1, \ldots, w_K)$ and let $\bar{w}$ be their average. For each $i$, set $z_i = \sqrt{K/(K-1)}\,(w_i - \bar{w})$. One can directly check that each $z_i$ is still marginally standard Gaussian, and $\mathrm{Corr}(z_{i,l}, z_{j,l}) = -1/(K-1)$ for every $i \neq j$ and every pixel $l$. In particular, when $K = 2$, this construction reduces to our usual antithetic noise pair.

**Quasi-Monte Carlo:** The idea of using negatively correlated samples has been widely adopted in Monte Carlo methods (Craiu & Meng, 2005) and statistical risk estimation (Liu et al., 2024) for improved performance. This principle of variance reduction can be further extended to QMC methods. As an alternative to Monte Carlo, QMC constructs a deterministic set of $N$ samples that have negative correlation and provide a more balanced coverage of the sample space. QMC samples can also be randomized, resulting in RQMC methods. RQMC maintains the marginal distribution of each sample while preserving the low-discrepancy properties of QMC. Importantly, repeated randomizations allow for empirical error estimation and confidence intervals (L'Ecuyer et al., 2023).

Under regularity conditions (Niederreiter, 1992), QMC and RQMC can improve the error *convergence rate* from $O(N^{-1/2})$ to $O(N^{-1+\epsilon})$ for any $\epsilon > 0$, albeit $\epsilon$ may absorb a dimension-dependent factor $(\log N)^d$. For sufficiently smooth functions, RQMC can achieve rates as fast as $O(N^{-3/2+\epsilon})$ (Owen, 1997b;a). Moreover, the space-filling property of QMC may also promote greater sample diversity.

Although the dimension of the Gaussian noise is higher than the typical regime where QMC is expected to be most effective, RQMC still further shrinks our confidence intervals by several-fold compared with standard MC. This hints that the image generator's effective dimension is much lower than its ambient one, allowing QMC methods to stay useful. Similar gains arise when QMC methods

deliberately exploit low-dimensional structure in practical problems (Wang & Fang, 2003; Xiao & Wang, 2019; Liu & Owen, 2023). Developing ways to identify and leverage this structure more systematically is an appealing avenue for future work.

# 5 EXPERIMENT

Sections 5.1 and 5.2 present two uncertainty quantification applications: estimating pixel-wise statistics and evaluating diffusion inverse problem solvers. The latter considers two popular algorithms across a range of tasks and datasets. For each task, we apply MC, AMC, and, when applicable, QMC estimators as described in Section 4. For each estimator, we report the $95\%$ confidence interval (CI) width and its relative efficiency. The relative efficiency of a new estimator (AMC or QMC) is defined as the squared ratio of the MC CI width to that of the estimator, $(\mathrm{CI_{MC}}/\mathrm{CI_{new\ estimator}})^2$. It reflects how many times more MC samples are required to achieve the same accuracy as our new estimator. Section 5.3 shows that antithetic noise produces more diverse images than independent noise while preserving quality, and Appendix C.2 explores an image editing example.

## 5.1 PIXEL-WISE STATISTICS

We begin by evaluating four pixel-level statistics. These include the (i) pixel value mean, (ii) perceived brightness, (iii) contrast, and (iv) the image centroid, with definitions in Appendix C.6. These statistics are actively used in diffusion workflows for diagnosing artifacts and assessing reliability, with applications in detecting signal leakage (Lin et al., 2024; Everaert et al., 2024), out-of-distribution detection (Le Bellier & Audebert, 2024), identifying artifact-prone regions (Kou et al., 2024), and improving the reliability of weather prediction (Li et al., 2024a).

**Setup:** We study both unconditional and conditional diffusion models; details are in Appendix C.3:

For *unconditional diffusion models*, we evaluate pre-trained models on CIFAR-10 (Krizhevsky et al., 2009), CelebA-HQ (Xia et al., 2021), and LSUN-Church (Yu et al., 2015). For each dataset, 1,600 image pairs are generated under both PN and RR noise sampling with 50 DDIM steps.

For *conditional diffusion models*, we evaluate Stable Diffusion 1.5 on 200 prompts from the Pick-a-Pic (Kirstain et al., 2023) and DrawBench (Saharia et al., 2022) with classifier-free-guidance (CFG) scale 3.5, and DiT (Peebles & Xie, 2023) on 32 ImageNet classes with CFG scale 4.0. For each class or prompt, 100 PN and RR pairs are generated with 20 DDIM steps. We also study the effect of CFG scale on both models in the Appendix C.4 (Ho & Salimans, 2022).

**Implementation:** To ensure fair comparisons under equal sample budget:

For *unconditional diffusion models*: MC uses 3,200 independent samples, AMC ($k = 2$) uses 1,600 antithetic pairs, and AMC ($k = 8$) uses 400 independent negative correlated batches. QMC employs a Sobol' point set of size 64 with 50 independent randomizations. Due to the dimensionality limits, QMC is restricted to CIFAR-10.

For *conditional diffusion models*: for each class or prompt, MC uses 200 independent samples, AMC uses 100 antithetic pairs, and QMC employs 8 Sobol' points with 25 randomizations.

CIs for MC and AMC are constructed as described in Section 4. For QMC, we use a student-$t$ interval to ensure reliable coverage (L'Ecuyer et al., 2023), with details in Appendix C.6.

**Results:** Across all statistics, both AMC and QMC have much shorter CIs than MC, with relative efficiencies ranging from 3.1 to 136. This strongly suggests our estimators can dramatically reduce cost for uncertainty estimates. AMC and QMC estimators yield comparable results. The performance of QMC depends on how the total budget is allocated between the size of the QMC point set and the number of random replicates, a trade-off we explore in further details in Appendix C.6.

## 5.2 EVALUATING DIFFERENT DIFFUSION INVERSE PROBLEM SOLVERS

Bayesian inverse problems are important in medical imaging, remote sensing, and astronomy. Recently, many methods have been proposed that leverage diffusion priors to solve inverse problems. For instance, Zheng et al. (2025) surveys 14 such methods, yet this is still far from exhaustive. As a

Table 2: CI lengths and efficiency $(\text{CI}_{\text{MC}}/\text{CI})^2$ (in parentheses), using MC as baseline.

| Dataset | | Brightness | Pixel Mean | Contrast | Centroid |
|---|---|---|---|---|---|
| CIFAR10 | MC | 2.00 | 2.04 | 1.08 | 0.14 |
| | QMC | 0.35 (32.05) | 0.39 (26.94) | **0.22 (23.43)** | 0.04 (9.65) |
| | AMC ($k = 2$) | **0.35 (32.66)** | 0.39 (27.12) | 0.23 (22.05) | 0.04 (9.73) |
| | AMC ($k = 8$) | 0.36 (30.35) | **0.39 (27.34)** | 0.24 (20.37) | **0.03 (13.94)** |
| CelebA | MC | 1.77 | 1.76 | 0.60 | 0.60 |
| | AMC ($k = 2$) | 0.26 (47.41) | 0.31 (33.15) | 0.19 (10.18) | 0.20 (7.82) |
| | AMC ($k = 8$) | **0.15 (130.69)** | **0.16 (115.79)** | **0.18 (10.90)** | **0.19 (8.49)** |
| Church | MC | 1.66 | 1.64 | 1.02 | 0.82 |
| | AMC ($k = 2$) | **0.14 (134.13)** | 0.16 (103.80) | 0.20 (27.16) | 0.26 (9.84) |
| | AMC ($k = 8$) | 0.15 (117.90) | **0.16 (107.64)** | **0.19 (27.81)** | **0.25 (10.87)** |
| Stable Diff. | MC | 3.86 | 3.95 | 2.18 | 4.03 |
| | QMC | 1.36 (8.07) | 1.43 (7.60) | 1.03 (4.46) | 1.88 (4.54) |
| | AMC ($k = 2$) | **0.90 (18.53)** | **1.00 (15.53)** | **0.89 (6.25)** | **1.62 (6.15)** |
| DiT | MC | 7.52 | 7.74 | 3.32 | 3.15 |
| | QMC | 3.44 (4.78) | 3.49 (4.91) | 1.82 (3.31) | **1.69 (3.46)** |
| | AMC ($k = 2$) | **3.13 (5.79)** | **3.12 (6.16)** | **1.72 (3.74)** | 1.74 (3.30) |

result, comprehensive benchmarking is costly. Meanwhile, most existing works on new diffusion inverse solvers (DIS) largely ignore uncertainty quantification, though it is critical for fair evaluation. This raises a key challenge: *how to efficiently evaluate posterior sampling algorithms with reliable estimates of reconstruction quality and uncertainty.* Our approach aims to efficiently address this gap.

We show that our antithetic estimator can save substantial computational cost for quantifying uncertainty by requiring far fewer samples than standard estimators. We focus on two popular DISs, Diffusion Posterior Sampling (DPS) (Chung et al., 2023a) and Decomposed Diffusion Sampler (DDS) (Chung et al., 2023b), and evaluate them on a range of tasks described below.

### 5.2.1 HUMAN FACE RECONSTRUCTION

We evaluate DPS across three inverse problems: inpainting, Gaussian deblurring, and super-resolution. Reconstruction error is measured using both PSNR and $L_1$ distance relative to the ground truth image. For all tasks, we use 200 human face images from the CelebA-HQ dataset. For each corrupted image, we generate 50 DPS reconstructions using both PN and RR noise pairs to compare their estimators. Operator-specific configurations (e.g., kernel sizes, mask ratios) are provided in Appendix C.6.

**Results:** As shown in Table 3, across all three tasks, AMC achieves substantially shorter confidence intervals than standard MC. This implies AMC yields efficiency gains, ranging from $54\% - 84\%$ in inpainting, $41\% - 54\%$ in super-resolution, and $34\% - 56\%$ in deblurring.

The reconstruction metrics ($L_1$, PSNR) differ by less than $0.2\%$ between the two estimators, showing that they produce consistent results. Meanwhile, our method offers a clear advantage in efficiency.

Table 3: Comparison of AMC vs. MC across tasks in DPS with efficiency $(\text{CI}_{\text{MC}}/\text{CI})^2$ in parentheses

| | L1 | | PSNR | |
|---|---|---|---|---|
| Task | AMC CI | MC CI | AMC CI | MC CI |
| Inpainting | **0.83 (1.54)** | 1.03 | **0.23 (1.84)** | 0.31 |
| Super-resolution | **1.01 (1.41)** | 1.20 | **0.24 (1.54)** | 0.30 |
| Gaussian Deblur | **0.89 (1.34)** | 1.03 | **0.23 (1.56)** | 0.29 |

### 5.2.2 MEDICAL IMAGE RECONSTRUCTION

We use the DDS (Chung et al., 2023b) as the reconstruction backbone and apply our antithetic noise initialization to estimate reconstruction confidence intervals. Given a single measurement, we generate 100 reconstruction pairs using 50-step DDS sampling. Reconstruction quality is evaluated against ground truth using $L_1$ distance and PSNR. Dataset details can be found in Appendix C.6.4.

**Results:** Both MC and AMC estimators give consistent estimates, with estimated $L_1$ error of 0.0194 and 0.0195, and PSNR values of 31.48 and 31.45, respectively. Similarly, we observe that AMC produces much shorter confidence intervals. The CI lengths decrease with relative efficiency of 1.44 for $L_1$ and 1.37 for PSNR. This shows that antithetic initialization consistently reduces estimator variance with no extra cost without degrading reconstruction fidelity in large-scale inverse problems.

**Conclusion:** These two experiments show that a single antithetic estimator reduces evaluation costs for diffusion inverse solvers by 34%–84%, a saving that becomes especially valuable given the large number of solvers in the literature. This makes large-scale benchmarking far more practical.

## 5.3 DIVERSITY IMPROVEMENT

Many diffusion tools generate a few images from one prompt to let users select a preferred one. However, these images may look similar with randomly sampled initial noise (Marwood et al., 2023; Sadat et al., 2024). We show that antithetic noise produces more diverse images than independent noise, without reducing image quality or increasing computational cost. Diversity is measured using pairwise SSIM (lower = more diverse) and LPIPS (Zhang et al., 2018) (higher = more diverse).

We evaluate the diversity metrics for both unconditional and conditional diffusion on image pairs generated following the same setup in Section 5.1. As shown in Table 4, antithetic noise pairs consistently lead to higher diversity, as indicated by lower SSIM and higher LPIPS scores. Importantly, image quality remains stable for both noise types. We expect our method can be easily integrated into existing diversity optimization techniques to further improve their performance.

Table 4: *Average percentage improvement* of PN pairs over RR pairs on SSIM and LPIPS.

| Metric | Unconditional Diffusion | | | Conditional Diffusion | |
|---|---|---|---|---|---|
| | CIFAR-10 | LSUN-Church | CelebA-HQ | SD1.5 | DiT |
| SSIM (%) | 88.78 | 45.69 | 36.78 | 28.32 | 23.99 |
| LPIPS (%) | 6.69 | 3.54 | 15.14 | 5.78 | 10.62 |

## 6 DISCUSSION AND FUTURE DIRECTIONS

We find that antithetic initial noise yields negatively correlated samples. This is a robust property that generalizes across every model that we have tested. The negative correlation is especially useful for uncertainty quantification, where it brings huge variance reduction and cost savings. This makes our approach highly effective for evaluation and benchmarking tasks. Meanwhile, this finding can serve as a simple plug-in to improve related tasks, such as diversity improvement and image editing.

A limitation of our work is that while the symmetry conjecture is supported by both empirical and theoretical evidence, it remains open in full generality. In addition, our method is not a cure-all: the LPIPS diversity improvements in Table 4 are consistent but modest. While we obtain large gains for pixel-wise statistics, the improvements for perceptual or text-alignment metrics such as MUSIQ and CLIP Score are small, since the strong nonlinearity of transformer decoders attenuates most of the negative correlation present in the diffusion outputs.

Looking forward, future directions include systematically studying the symmetry conjecture; integrating antithetic noise with existing noise-optimization methods; and further leveraging QMC to improve sampling and estimation.

## ACKNOWLEDGEMENT

Guanyang Wang and Jing Jia acknowledge support from the National Science Foundation through grant DMS–2210849 and an Adobe Data Science Research Award. Liyue Shen and Bowen Song acknowledge funding support from the National Science Foundation through grant IIS-2435746, Defense Advanced Research Projects Agency (DARPA) under Contract No. HR00112520042, Hyundai America Technical Center, Inc. (HATCI), as well as the University of Michigan MICDE Catalyst Grant Award and MIDAS PODS Grant Award.

## ETHICS STATEMENT

This work does not involve human subjects or personally identifiable data collected by the authors. All datasets used are publicly available and distributed under licenses permitting academic research.

Our medical image reconstruction experiments rely on benchmark datasets that are fully de-identified before release (e.g., publicly available MRI data). No new data collection was performed, and no protected health information is included. The experiments are purely methodological, intended to evaluate reconstruction fidelity and uncertainty quantification techniques, and make no clinical claims or diagnostic recommendations.

Our contributions are methodological, focusing on sampling and uncertainty quantification in generative models. While such methods could be applied broadly, including in sensitive domains, we emphasize that our work aims to improve reliability, transparency, and efficiency in evaluation.

## REPRODUCIBILITY STATEMENT

Full experimental details can be found in Section 5 and Appendix C. The code is available at `https://github.com/jjia131/Antithetic-Noise-in-Diffusion-Models-page`.

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

## A   USE OF LLMs

LLMs are used only for writing, editing, or formatting purposes and do not affect the core methodology, contributions, and originality of the paper.

## B   PROOFS

### B.1   PROOFS IN SECTION 3.2

*Proof of Lemma 1.*  It suffices to show $\mathrm{Var}(f(Z) + f(-Z)) = 0$. By definition:

$$\mathrm{Corr}(f(-Z), f(Z)) = \frac{\mathrm{Cov}(f(-Z), f(Z))}{\sqrt{\mathrm{Var}(f(Z))\,\mathrm{Var}(f(-Z))}} = -1.$$

Therefore

$$\begin{aligned}
\mathrm{Var}(f(Z) + f(-Z)) &= \mathrm{Var}(f(Z)) + \mathrm{Var}(f(-Z)) + 2\,\mathrm{Cov}(f(Z), f(-Z)) \\
&= \mathrm{Var}(f(Z)) + \mathrm{Var}(f(-Z)) - 2\sqrt{\mathrm{Var}(f(Z))\,\mathrm{Var}(f(-Z))} \\
&= \left(\sqrt{\mathrm{Var}(f(Z))} - \sqrt{\mathrm{Var}(f(-Z))}\right)^2.
\end{aligned}$$

On the other hand, since $-Z \sim \mathbb{N}(0, I)$ given $Z \sim \mathbb{N}(0, 1)$, we have $\mathrm{Var}(f(Z)) = \mathrm{Var}(f(-Z))$ and then $\mathrm{Var}(f(Z) + f(-Z)) = 0$, as claimed.  □

### B.2   THEORETICAL GUARANTEE OF THE SYMMETRY CONJECTURE FOR LARGE $t$

**Background: The Ornstein–Uhlenbeck (OU) process.**    The OU process $(X_t)_{t \geq 0}$ in $\mathbb{R}^d$ solves the SDE

$$\mathrm{d}X_t = -X_t\,\mathrm{d}t + \sqrt{2}\,\mathrm{d}B_t, \qquad X_0 \sim \mu_0,$$

where $(B_t)$ is standard Brownian motion in $\mathbb{R}^d$. This is a Gaussian Markov process with stationary distribution $\gamma_d = \mathbb{N}(0, I_d)$ and generator

$$Lf(x) = \Delta f(x) - x \cdot \nabla f(x).$$

For any initial law $\mu_0$, let $\mu_t$ denote the law of $X_t$. Then $\mu_t = \mu_0 P_t$, where $(P_t)_{t \geq 0}$ is the *OU semigroup* defined by

$$(P_t f)(x) = \mathbb{E}[f(X_t) \mid X_0 = x].$$

Write the *score* of $\mu_t$ as $s_t(x) := \nabla \log p_t(x)$, where $p_t$ is Lebesgue density of $\mu_t$, we have

**Theorem 1** (Score converges to the Gaussian score)**.**  *Under the setup above,*

$$\lim_{t \to \infty} \mathbb{E}_{\mu_t}\big[\|s_t(X_t) + X_t\|^2\big] = 0.$$

*Moreover, the convergence is quantified by*

$$\mathbb{E}_{\mu_t}\big[\|s_t(X_t) + X_t\|^2\big] = \mathcal{I}(\mu_t \mid \gamma_d) \leq e^{-2t}\,\mathcal{I}(\mu_0 \mid \gamma_d).$$

*Proof.*  Recall the definition of relative Fisher information

$$\mathcal{I}(\mu_t \mid \gamma_d) := \mathbb{E}_{\mu_t}\big[\|\nabla \log(p_t/\gamma_d)\|^2\big].$$

Since $\nabla \gamma_d(x) = -x$, we have:

$$\mathcal{I}(\mu_t \mid \gamma_d) = \mathbb{E}_{\mu_t}\big[\|s_t(X_t) + X_t\|^2\big].$$

Moreover, it is well known (e.g. Chapter 5 of Bakry et al. (2013)) that $\mathcal{I}(\mu_t \mid \gamma_d) \leq \exp(-2t)\mathcal{I}(\mu_0 \mid \gamma_d)$, this concludes our proof.  □

In variance-preserving diffusion models, the forward noising process is exactly the Ornstein–Uhlenbeck semigroup. Theorem 1 implies that the true score converges to the Gaussian score $-x$. Consequently, one-dimensional slices of the score become asymptotically affine antisymmetric, confirming our symmetry conjecture in the high-noise limit.

The next corollary shows, the 1-step DDIM update has nearly $-1$ correlation at large $t$.

**Corollary 1** (Correlation of 1-step DDIM). *With all the setup the same as above, assuming we have a score network $\epsilon^{(t)}$ approximating the score function. Let $\eta_t := \max\{\mathbb{E}_{p_t}\|\epsilon^{(t)}(X) - s_t(X)\|^2, \mathbb{E}_{p_t}\|\epsilon^{(t)}(-X) - s_t(-X)\|^2\}$ be the expected (symmetrized) error. Consider one-step DDIM update $F_t(x) = a_t x + b_t\,\epsilon_\theta^{(t)}(x) := (F_{t,1}(x), \dots, F_{t,d}(x)) \in \mathbb{R}^d$ as defined in (2). Let*

$$v_{t,i} := \min\{\mathrm{Var}_{p_t}\big(F_{t,i}(X)\big), \mathrm{Var}_{p_t}\big(F_{t,i}(-X)\big)\}.$$

*Then*

$$\big|\mathrm{Corr}\big(F_{t,i}(X), F_{t,i}(-X)\big) + 1\big| \;\leq\; \frac{2\,|b_t|}{\sqrt{v_{t,i}}}\Big(\sqrt{\eta_t} + e^{-t}\sqrt{\mathcal{I}(\mu_0 \mid \gamma_d)}\Big).$$

*Proof.* Theorem 1 shows $\mathcal{I}(\mu_t \mid \gamma_d) \leq \exp(-2t)\mathcal{I}(\mu_0 \mid \gamma_d)$. Write $\epsilon_\theta^{(t)} = s_t + r_t$ and $s_t = -x + \Delta_t$, so that $\mathbb{E}_{p_t}\|r_t(X)\|^2 \leq \eta_t$ and $\mathbb{E}_{p_t}\|\Delta_t(X)\|^2 = \mathcal{I}(\mu_t \mid \gamma_d)$. Then

$$F_t(x) = (a_t - b_t)x + b_t\big(\Delta_t(x) + r_t(x)\big), \qquad F_t(-x) = -F_t(x) + E_t(x),$$

with $E_t(x) := b_t[\Delta_t(-x) + \Delta_t(x) + r_t(-x) + r_t(x)]$. By triangle inequality of $L^2(p_t)$ norm,

$$\|E_t\|_{L^2(p_t)} \;\leq\; |b_t|\big(e^{-t}\sqrt{\mathcal{I}(\mu_0 \mid \gamma_d)} + e^{-t}\sqrt{\mathcal{I}(\mu_0 \mid \gamma_d)} + 2\sqrt{\eta_t}\big).$$

Using

$$\mathrm{Corr}\big(F_{t,i}(X), F_{t,i}(-X)\big) = -1 + \frac{\mathrm{Cov}\big(F_{t,i}(X), E_{t,i}(X)\big)}{\sqrt{\mathrm{Var}(F_{t,i}(X))\,\mathrm{Var}(F_{t,i}(-X))}}$$

and Cauchy–Schwarz with the variance lower bounds gives

$$\big|\mathrm{Corr}\big(F_t(X), F_t(-X)\big) + 1\big| \;\leq\; \frac{\|E_t\|_{L^2(p_t)}}{\sqrt{v_{t,i}}} \;\leq\; \frac{2\,|b_t|}{\sqrt{v_{t,i}}}\big(\sqrt{\eta_t} + e^{-t}\sqrt{\mathcal{I}(\mu_0 \mid \gamma_d)},$$

as claimed. $\qquad\square$

In practice, if $v_{t,i} > 0$, the correlation is close to $-1$ for large $t$; its deviation from $-1$ is on the order of the neural network approximation error plus an exponentially decaying term $Ce^{-t}$.

### B.3 MONOTONE CONVERGENCE IN $t$

Section B.2 shows that the score converges to the Gaussian score as $t \to \infty$. We now further prove that (1) the score error $\mathbb{E}_{\mu_t}\big[|s_t(X_t) + X_t|^2\big]$ and (2) the density ratio $p_t/\gamma_d$ both converge monotonically in $t$.

#### B.3.1 DENSITY RATIO CONVERGENCE

**Background: Hermite Polynomials in $\mathbb{R}$:** For $x \in \mathbb{R}$, the *(probabilist's) Hermite polynomials* $\{H_n(x)\}_{n\geq 0}$ form a sequence of polynomials defined by

$$H_n(x) = (-1)^n \exp\left(\frac{x^2}{2}\right) \frac{\mathrm{d}^n}{\mathrm{d}x^n} \exp\left(-\frac{x^2}{2}\right).$$

We summarize their known properties here. The first two can be checked by direct calculation. The latter two can be found on page 105 of Bakry et al. (2013).

**Proposition 1.** *Hermite polynomials $\{H_n(x)\}_{n\geq 0}$ satisfy the following*

- *(Recurrence relation)* $H_{n+1}(x) = x\,H_n(x) - H'_n(x), \qquad n \geq 0.$

- *(Orthogonality under Gaussian measure) We have:*

$$\int H_n(x) H_m(x) \gamma_1(\mathrm{d}x) = n! \delta_{nm},$$

  *where $\gamma_1(\mathrm{d}x) = (2\pi)^{-1/2} e^{-x^2/2} \mathrm{d}x$ denotes the one-dimensional standard Gaussian measure, and $\delta_{nm}$ is 1 when $n = m$ and 0 otherwise.*

- *(Completeness) Hermite polynomials $\{H_n(x)\}_{n \geq 0}$ form an orthogonal basis of the Hilbert space $L^2(\gamma) := \{f : \int f^2 \gamma(\mathrm{d}x) < \infty\}$ equipped with inner product $\langle f, g \rangle_\gamma := \int f g \gamma(\mathrm{d}x)$.*

- *(Eigenfunction) Let $L$ be a differential operator defined as $L(f) := f'' - x f'$. Hermite polynomials $\{H_n(x)\}_{n \geq 0}$ are eigenfunctions of $L$:*

$$L(H_n) = -n H_n, \qquad n \geq 0.$$

**Hermite Polynomials in $\mathbb{R}^d$:** One can naturally extend univariate Hermite polynomials to the multivariate setting. For $x = (x_1, \ldots, x_d) \in \mathbb{R}^d$, let $\gamma_d$ denote the $d$-dimensional standard Gaussian measure,

$$\gamma_d(\mathrm{d}x) := (2\pi)^{-d/2} \exp\left(-\frac{\|x\|^2}{2}\right) \mathrm{d}x.$$

For a multi-index $\alpha = (\alpha_1, \ldots, \alpha_d) \in \mathbb{N}^d$, we define the *multivariate Hermite polynomial*

$$H_\alpha(x) := \prod_{i=1}^d H_{\alpha_i}(x_i),$$

where $H_{\alpha_i}$ is the one-dimensional probabilists' Hermite polynomial. We write $|\alpha| := \alpha_1 + \cdots + \alpha_d$ and $\alpha! := \alpha_1! \cdots \alpha_d!$. Similar to the one-dimensional case, the family $\{H_\alpha\}_{\alpha \in \mathcal{N}^d}$ forms an orthogonal basis of the Hilbert space $L^2(\gamma_d)$:

$$\int_{\mathbb{R}^d} H_\alpha(x) \, H_\beta(x) \, \gamma_d(\mathrm{d}x) = \alpha! \, \delta_{\alpha\beta}.$$

**Monotone convergence of the density ratio.** Recall from Section B.2 that the generator of the OU process

$$Lf(x) = \Delta f(x) - x \cdot \nabla f(x).$$

and the corresponding Markov semigroup $P_t = \exp(tL)$. We show that the multivariate Hermite polynomials are eigenfunctions of $L$, and thus also of $P_t$.

**Proposition 2.** *We have:*

- $LH_\alpha = -|\alpha| H_\alpha, \qquad \alpha \in \mathbb{N}^d,$

- $P_t H_\alpha = e^{-|\alpha|t} H_\alpha, \qquad \alpha \in \mathbb{N}^d, t \geq 0.$

- *For every $g \in L^2(\gamma_d)$*

$$P_t g(x) = \sum_{\alpha \in \mathbb{N}^d} a_\alpha e^{-t|\alpha|} H_\alpha(x),$$

  *where $a_\alpha = \langle g, H_\alpha \rangle_{L^2(\gamma_d)}$.*

*Proof.* We can directly calculate

$$
\begin{aligned}
LH_\alpha(x) &= \Delta H_\alpha(x) - x \cdot \nabla H_\alpha(x) \\
&= \sum_i H_{\alpha_i}''(x_i) H_{\alpha_{-i}}(x_{-i}) - \sum_i x_i H_{\alpha_i}'(x_i) H_{\alpha_{-i}}(x_{-i}) \\
&= \sum_i H_{\alpha_{-i}}(x_{-i})(H_{\alpha_i}''(x_i) - x_i H_{\alpha_i}'(x_i)) \\
&= \sum_i (-\alpha_i) H_{\alpha_{-i}}(x_{-i}) H_{\alpha_i}(x_i) \\
&= -|\alpha| H_\alpha(x),
\end{aligned}
$$

where $H_{\alpha_{-i}}(x_{-i}) = \prod_{j \neq i} H_{\alpha_j}(x_j)$. The second to last equality follows from the fourth property of Proposition 1.

Consequently, the OU semigroup $(P_t)_{t \geq 0}$ satisfies

$$P_t H_\alpha = e^{-t|\alpha|} H_\alpha, \qquad t \geq 0.$$

For every $g \in L^2(\gamma_d)$, we rewrite $g$ according to its Hermite expansion (which is valid according to properties 2-3 in Proposition 1).

$$g(x) = \sum_{\alpha \in \mathbb{N}^d} a_\alpha H_\alpha(x), \qquad a_\alpha = \langle g, H_\alpha \rangle_{L^2(\gamma_d)}.$$

Therefore

$$P_t g(x) = \sum_{\alpha \in \mathbb{N}^d} a_\alpha e^{-t|\alpha|} H_\alpha(x),$$

as claimed. $\qquad \square$

In particular, let $p_t(x) = f_t(x)\gamma_d(x)$ denote the density at time $t$ of the OU process started from $p_0$. It is known that $f_t = P_t f_0$. Proposition 2 implies

$$f_t(x) = \sum_{\alpha \in \mathbb{N}^d} a_\alpha e^{-t|\alpha|} H_\alpha(x) = a_0 + \sum_{|\alpha| > 0} a_\alpha e^{-t|\alpha|} H_\alpha(x),$$

where $a_\alpha$ are the Hermite coefficients of $f_0$. Since $H_0 \equiv 1$ and

$$a_0 = \int_{\mathbb{R}^d} f_0(x) \, H_0(x) \, \gamma_d(\mathrm{d}x) = \int_{\mathbb{R}^d} f_0(x) \, \gamma_d(\mathrm{d}x) = \int_{\mathbb{R}^d} p_0(x) \, \mathrm{d}x = 1,$$

we may rewrite

$$f_t(x) = 1 + \sum_{|\alpha| > 0} a_\alpha e^{-t|\alpha|} H_\alpha(x). \tag{3}$$

We now show $f_t$ monotonically converges to 1 in $L^2(\gamma_d)$.

**Proposition 3.** *Let $p_t = f_t \gamma_d$ be as above, and assume $f_0 \in L^2(\gamma_d)$. Then $\|f_t - 1\|_{L^2(\gamma_d)}$ monotonically decays to 0 at the speed of $e^{-t}$.*

*Proof.* From (3) we have

$$f_t(x) = 1 + \sum_{|\alpha| > 0} a_\alpha e^{-t|\alpha|} H_\alpha(x).$$

Since $\{H_\alpha\}_{\alpha \in \mathbb{N}^d}$ is an orthogonal basis of $L^2(\gamma_d)$, we obtain

$$\|f_t - 1\|_{L^2(\gamma_d)}^2 = \sum_{\alpha \in \mathbb{N}^d} a_\alpha^2 e^{-2t|\alpha|}.$$

Since each term in the sum decreases monotonically to 0, the same holds for $\|f_t - 1\|_{L^2(\gamma_d)}^2$. The overall decay rate is governed by the slowest mode, which corresponds to the smallest $|a| = 1$, thus the rate of convergence is $\exp(-2t)$. Taking square roots yields the desired result. $\qquad \square$

### B.3.2 Score Error Convergence

Recall from Section B.2 that $\mathbb{E}_{\mu_t}\big[\|s_t(X_t) + X_t\|^2\big] = \mathcal{I}(\mu_t \mid \gamma_d)$. It follows from equation 5.7.2 in Bakry et al. (2013) that

$$\frac{\mathrm{d}\mathcal{I}(\mu_t \mid \gamma_d)}{\mathrm{d}t} \leq -2\mathcal{I}(\mu_t \mid \gamma_d) \leq 0.$$

Thus $\mathcal{I}(\mu_t \mid \gamma_d)$ is monotonically decreasing as a function of $t$. Further, Grönwall's inequality implies

$$\mathcal{I}(\mu_t \mid \gamma_d) \leq e^{-2t} \mathcal{I}(\mu_0 \mid \gamma_d).$$

## B.4 SYMMETRY PRESERVATION UNDER THE FORWARD PROCESS

This section shows the forward process preserves any orthogonal symmetry in the data distribution. More precisely, let $G$ be a subgroup of the orthogonal group $O(d)$ in $\mathbb{R}^d$ (e.g., coordinate sign flips, coordinate permutations, and rotations). We have the following:

**Proposition 4.** *Assuming $p_0$ is symmetric about $\mu$ under the action of $G$, i.e., for every $g \in G$, we have $p_0(x) = p_0(g \cdot (x - \mu) + \mu)$ for every $x$. Then $p_t$ satisfies $p_t(x) = p_t(g \cdot (x - \mu_t) + \mu_t)$ for every $x$, and $s_t(\mu_t + g \cdot x) = g \cdot s_t(\mu_t + x)$, where $\mu_t = e^{-t}\mu$.*

*Proof.* Let $X_0 \sim p_0, Z \sim \mathbb{N}(0, I_d)$. It is known that $X_t := e^{-t}X_0 + \sqrt{1 - e^{-2t}}Z \sim p_t$. For any $g \in G$,

$$g \cdot (X_t - \mu_t) + \mu_t = g \cdot (e^{-t}X_0 + \sqrt{1 - e^{-2t}}Z - \mu_t) + \mu_t$$
$$= e^{-t}(g \cdot (X_0 - \mu) + \mu) + \sqrt{1 - e^{-2t}}g \cdot Z \sim p_t$$

where the last claim uses that $g \cdot (X_0 - \mu) + \mu$ has the same distribution as $X_0$ (by symmetry) and $g \cdot Z \sim \mathbb{N}(0, I_d)$. This proves the claim on the symmetry of $p_t$. For the score, since $p_t(\mu_t + g \cdot x) = p_t(\mu_t + x)$, taking derivative on both sides yields:

$$g^\top \cdot \nabla p_t(\mu_t + g \cdot x) = \nabla p_t(\mu_t + x).$$

Since $g$ is an orthogonal matrix, $g^\top = g^{-1}$, we have:

$$\nabla p_t(\mu_t + g \cdot x) = g \cdot \nabla p_t(\mu_t + x).$$

Dividing both sides by $p_t(\mu_t + x)$ proves the claimed result. $\square$

Sections B.2 (large $t$ limit), B.3 (monotone convergence), and B.4 (symmetry perservation from $t = 0$) together provide insights for the conjecture: The forward OU process preserves all orthogonal symmetries of $p_0$, including central reflections. Hence, in the idealized case where $p_0$ is reflection-symmetric, the density is even and its score is odd. Proposition 4 ensures that this symmetry is preserved for all $t$. Furthermore, even when the symmetry is only approximate near $t \approx 0$, the monotone convergence of $p_t$ toward the Gaussian is exponentially fast, so the forward dynamics push $s_t$ toward a linear function in a monotone manner.

## B.5 PROOFS IN SECTION 4

### B.5.1 MONTE CARLO ESTIMATOR:

**Proposition 5.** *Denote $\mathbb{E}_{z \sim \mathbb{N}(0,I)}[S(DM(z))]$ by $\mu$ and $\text{Var}_{z \sim \mathbb{N}(0,I)}[S(DM(z))]$ by $\sigma^2$. Assuming $\sigma^2 < \infty$, then we have:*

- $\hat{\mu}_N^{MC} \to \mu$ and $(\hat{\sigma}_N^{MC})^2 \to \sigma^2$ almost surely, and $\mathbb{P}\left(\mu \in CI_N^{MC}(1 - \alpha)\right) \to 1 - \alpha$, both as $N \to \infty$.

- $\mathbb{E}[(\hat{\mu}_N^{MC} - \mu)^2] = \text{Var}[\hat{\mu}_N^{MC}] = \sigma^2/N.$

In words, the above proposition shows: (i) *Correctness*: The standard Monte Carlo estimator $\hat{\mu}_N^{\text{MC}}$ converges to the true value, and the coverage probability of $\text{CI}_N^{\text{MC}}(1 - \alpha)$ converges to the nominal level $(1 - \alpha)$. (ii) *Reliability*: The expected squared error of $\hat{\mu}_N^{\text{MC}}$ equals the variance of a single sample divided by sample size. The confidence interval has width approximately $2\sigma z_{1-\alpha/2}/\sqrt{N}$.

*Proof of Proposition 5.* Let $S_i := S(DM(z_i))$ for $i = 1, \ldots, N$. Because the noises $z_i$ are drawn independently from $\mathbb{N}(0, I)$, the random variables $S_1, S_2, \ldots$ are independent and identically distributed with mean $\mu$ and variance $\sigma^2 < \infty$.

**Consistency.** By the *strong law of large numbers*,

$$\hat{\mu}_N^{\text{MC}} = \frac{1}{N}\sum_{i=1}^{N} S_i \xrightarrow{\text{a.s.}} \mu.$$

The sample variance estimator

$$(\hat{\sigma}_N^{\text{MC}})^2 = \frac{1}{N-1}\sum_{i=1}^{N}(S_i - \hat{\mu}_N^{\text{MC}})^2 = \frac{1}{N-1}\sum_{i=1}^{N} S_i^2 - \frac{N}{N-1}(\hat{\mu}_N^{\text{MC}})^2.$$

The first term converges to $\mathbb{E}[S_1^2]$ almost surely by the law of large numbers, the second term converges to $\mu^2$ almost surely as shown above. Therefore $(\hat{\sigma}_N^{\text{MC}})^2 \xrightarrow{\text{a.s.}} \sigma^2$.

**Asymptotic normality and coverage.** The classical *central limit theorem* states that

$$\sqrt{N}\,\frac{\hat{\mu}_N^{\text{MC}} - \mu}{\sigma} \xrightarrow{d} \mathbb{N}(0,1).$$

Replacing the unknown $\sigma$ by the consistent estimator $\hat{\sigma}_N^{\text{MC}}$ and applying *Slutsky's theorem* yields

$$\sqrt{N}\,\frac{\hat{\mu}_N^{\text{MC}} - \mu}{\hat{\sigma}_N^{\text{MC}}} \xrightarrow{d} \mathbb{N}(0,1).$$

Hence

$$\mathbb{P}\big(\mu \in \text{CI}_N^{\text{MC}}(1-\alpha)\big) = \mathbb{P}\Big(\big|\sqrt{N}(\hat{\mu}_N^{\text{MC}} - \mu)/\hat{\sigma}_N^{\text{MC}}\big| \le z_{1-\alpha/2}\Big) \longrightarrow 1 - \alpha.$$

**Mean-squared error.** Because $\hat{\mu}_N^{\text{MC}}$ is the average of $N$ i.i.d. variables,

$$\text{Var}\big[\hat{\mu}_N^{\text{MC}}\big] = \frac{\sigma^2}{N}.$$

Moreover, since $\hat{\mu}_N^{\text{MC}}$ is unbiased ($\mathbb{E}\big[\hat{\mu}_N^{\text{MC}}\big] = \mu$), its mean-squared error

$$\mathbb{E}\big[(\hat{\mu}_N^{\text{MC}} - \mu)^2\big] = \mathbb{E}\big[(\hat{\mu}_N^{\text{MC}} - \mathbb{E}[\hat{\mu}_N^{\text{MC}}])^2\big] = \frac{\sigma^2}{N}.$$

This completes the proof. $\qquad\qquad\qquad\qquad\qquad\qquad\qquad\qquad\qquad\qquad\qquad\square$

### B.5.2 ANTITHETIC MONTE CARLO ESTIMATOR:

**Proposition 6.** *Denote $\mathbb{E}[S(DM(z))]$ by $\mu$, $\text{Var}[S(DM(z))]$ by $\sigma^2$, and $\text{Cov}(S_1^+, S_1^-)$ by $\rho$. Assuming $\sigma^2 < \infty$, then we have:*

- *$\hat{\mu}_N^{AMC} \to \mu$ and $(\hat{\sigma}_N^{AMC})^2 \to (1+\rho)\sigma^2/2$ almost surely as $N \to \infty$,*

- *$\mathbb{E}[(\mu_N^{AMC} - \mu)^2] = \text{Var}[\mu_N^{AMC}] = \sigma^2(1+\rho)/N$.*

- *$\mathbb{P}\big(\mu \in CI_N^{AMC}(1-\alpha)\big) \to 1 - \alpha$ as $N \to \infty$.*

*Proof.* Let $N = 2K$ and generate independent antithetic noise pairs $(z_i, -z_i)$ for $i = 1,\dots,K$. Recall

$$S_i^+ = S\big(\text{DM}(z_i)\big), \quad S_i^- = S\big(\text{DM}(-z_i)\big), \quad \bar{S}_i = \tfrac{1}{2}\big(S_i^+ + S_i^-\big), \quad i = 1,\dots,K.$$

Because the pairs are independent and identically distributed (i.i.d.), the random variables $\bar{S}_1,\dots,\bar{S}_K$ are i.i.d. with

$$\mathbb{E}[\bar{S}_1] = \mu$$

and

$$\text{Var}[\bar{S}_1] = \text{Var}\left[\tfrac{1}{2}\left(S_1^+ + S_1^-\right)\right] \tag{4}$$

$$= \left(\tfrac{1}{2}\right)^2 \text{Var}\left[S_i^+ + S_i^-\right] \tag{5}$$

$$= \frac{1}{4}\left(\text{Var}[S_i^+] + \text{Var}[S_i^-] + 2\,\text{Cov}(S_i^+, S_i^-)\right) \tag{6}$$

$$= \frac{1}{4}\left(\sigma^2 + \sigma^2 + 2\rho\sigma^2\right) \tag{7}$$

$$= \frac{1+\rho}{2}\,\sigma^2. \tag{8}$$

where we have written $\text{Cov}(S_i^+, S_i^-) = \rho\,\sigma^2$ as in the statement. Set $\hat{\mu}_N^{\text{AMC}} = K^{-1}\sum_{i=1}^K \bar{S}_i$ and let $(\hat{\sigma}_N^{\text{AMC}})^2$ be the sample variance of $\bar{S}_1, \ldots, \bar{S}_K$. All three claims in Proposition 6 follow from Proposition 5.

$\square$

## B.6 An alternative explanation via the FKG inequality

We also provide an expository discussion highlighting the FKG connection behind negative correlation. Consider the univariate case where a scalar Gaussian $z$ is fed through a sequence of one-dimensional linear maps and ReLU activations. Let $F$ be the resulting composite function. One can show that, regardless of the number of layers or the linear coefficients used, $\text{Corr}\big(F(z),\,F(-z)\big) \leq 0$. The proof relies on the univariate FKG inequality (Fortuin et al., 1971), a well-known result in statistical physics and probability theory.

We generalize this result to higher dimensions via *partial monotonicity*, under which negative correlation still holds.

We first formally state and prove the claim for the univariate case in Section B.6.

**Proposition 7** (Univariate case). *Let $z \sim \mathbb{N}(0,1)$ and let $F : \mathbb{R} \to \mathbb{R}$ be the output of any one–dimensional feed-forward network obtained by alternating scalar linear maps and ReLU activations:*

$$h_0(z) = z, \qquad h_\ell(z) = \text{ReLU}\big(w_\ell\,h_{\ell-1}(z) + b_\ell\big), \ \ \ell = 1, \ldots, L, \qquad F(z) = h_L(z).$$

*For all choices of depths $L$ and coefficients $\{w_\ell, b_\ell\}_{\ell=1}^L$,*

$$\text{Corr}\big(F(z),\,F(-z)\big) \ \leq \ 0.$$

*Proof.* An important observation is that monotonicity is preserved under composition: combining one monotonic function with another produces a function that remains monotonic.

Each scalar linear map $x \mapsto w_\ell x + b_\ell$ is monotone: it is non-decreasing if $w_\ell \geq 0$ and non-increasing if $w_\ell < 0$. The ReLU map $x \mapsto \text{ReLU}(x) = \max\{0, x\}$ is non-decreasing. Hence the final function $F$ is monotone. Without loss of generality, we assume $F$ is non-decreasing.

FKG inequality guarantees $\text{Cov}_{Z\sim\mathbb{N}(0,1)}(f(Z), g(Z)) \geq 0$ provided that $f, g$ are non-decreasing. Therefore, $\text{Cov}_{Z\sim\mathbb{N}(0,1)}(F(Z), F(-Z)) = -\,\text{Cov}_{Z\sim\mathbb{N}(0,1)}(F(Z), -F(-Z)) \leq 0$. Since

$$\text{Corr}\big(F(z),\,F(-z)\big) = \frac{\text{Cov}_{Z\sim\mathbb{N}(0,1)}(F(Z), F(-Z))}{\sqrt{\text{Var}\big(F(z)\big)\,\text{Var}\big(F(-z)\big)}},$$

we have $\text{Corr}\big(F(z),\,F(-z)\big) \leq 0$. $\square$

To generalize the result to higher dimension, we define a function $f : \mathbb{R}^m \to \mathbb{R}$ to be *partially monotone* if for each coordinate $j$, holding all other inputs fixed, the map $t \mapsto f(x_1, \ldots, x_{j-1}, t, x_{j+1}, \ldots, x_m)$ is either non-decreasing or non-increasing. Mixed monotonicity is allowed. For example, $f(x, y) = x - y$ is non-decreasing in $x$ and non-increasing in $y$, yet still qualifies as partially monotone. We have the following result:

**Proposition 8.** *For a diffusion model $DM : \mathbb{R}^d \to \mathbb{R}^m$ and a summary statistics $S : \mathbb{R}^m \to \mathbb{R}$, if the joint map is partially monotone, then $\mathrm{Corr}(S \circ DM(Z)), S \circ DM(-Z)) \leq 0$.*

Now we prove the general case:

*Proof of Proposition 8.* Let $G := S \circ \mathrm{DM}$. For each coordinate $j \in [d]$ fix a sign

$$s_j = \begin{cases} +1, & \text{if } G \text{ is non–decreasing in } x_j, \\ -1, & \text{if } G \text{ is non–increasing in } x_j. \end{cases}$$

Write $s = (s_1, \ldots, s_d) \in \{\pm 1\}^d$ and define, for any $z \in \mathbb{R}^d$,

$$\widetilde{G}(z) = \widetilde{G}(z_1, \ldots, z_d) := G(s_1 z_1, \ldots, s_d z_d).$$

Similarly, define

$$\widetilde{H}(z) := \widetilde{G}(-z) = G(-s_1 z_1, \ldots, -s_d z_d).$$

By construction, $\widetilde{G}$ is coordinate-wise non–decreasing and $\widetilde{H}$ is coordinate-wise non–increasing.

Because each coordinate of $Z \sim \mathbb{N}(0, I_d)$ is symmetric, the random vectors $(s_1 Z_1, \ldots, s_d Z_d)$ and $(Z_1, \ldots, Z_d)$ have the same distribution. Hence

$$\mathrm{Cov}\big(G(Z), G(-Z)\big) = \mathrm{Cov}\big(\widetilde{G}(Z), \widetilde{H}(Z)\big).$$

The multivariate FKG inequality (Fortuin et al., 1971) for product Gaussian measures states that, when $U$ and $V$ are coordinate-wise non–decreasing, $\mathrm{Cov}\big(U(Z), V(Z)\big) \geq 0$. Apply it to the pair $\big(\widetilde{G}, -\widetilde{H}\big)$: both components are non–decreasing, so

$$\mathrm{Cov}\big(\widetilde{G}(Z), -\widetilde{H}(Z)\big) \geq 0 \implies \mathrm{Cov}\big(\widetilde{G}(Z), \widetilde{H}(Z)\big) \leq 0 \Leftrightarrow \mathrm{Cov}\big(G(Z), G(-Z)\big) \leq 0.$$

Therefore $\mathrm{Corr}\big(G(Z), G(-Z)\big) \leq 0$, as claimed. $\square$

Proposition 8 relies on checking partial-monotonicity. If $S$ is partially monotone (including any linear statistic), then the conditions of Proposition 8 are satisfied by, e.g., Neural Additive Models (Agarwal et al., 2021) and Deep Lattice Networks (You et al., 2017). Unfortunately, partial-monotonicity is in general hard to verify.

While popular diffusion architectures like DiT and U-Net lack partial monotonicity, we include Proposition 8 as an expository attempt to highlight the FKG connection behind negative correlation.

### B.6.1 FKG INEQUALITY FOR DDIM

We first analyze the idealized DDIM process using the FKG inequality. A single step of the idealized DDIM is

$$F_t(\mathbf{x}) = a_t \mathbf{x} + c_t s_t(\mathbf{x}), \tag{9}$$

where $s_t$ denotes the score of $p_t$, and $a_t, c_t \geq 0$ are deterministic coefficients obtained by rearranging (2).

The idealized DDIM trajectory is given by the composition:

$$\mathrm{DDIM_I} = F_1 \circ F_2 \circ F_3 \circ \cdots \circ F_T. \tag{10}$$

We present two results that give sufficient conditions under which the local (one-step) DDIM update (9) and the global DDIM procedure (10) generate negative correlation.

We need the following definition:

**Definition 1** (MTP$_2$ with curvature bound). *Let $p : \mathbb{R}^d \to (0, \infty)$ be a $C^2$ probability density. We say that $p$ is multivariate totally positive of order 2 (MTP$_2$) with curvature bound $\kappa \geq 0$ if the second partial derivatives of $\log p$ satisfy, for all $x \in \mathbb{R}^d$,*

$$\partial_{ij}^2 \log p(x) \geq 0 \quad \text{for all } i \neq j, \qquad \text{and} \qquad \partial_{ii}^2 \log p(x) \geq -\kappa \quad \text{for all } i.$$

$\text{MTP}_2$ distributions are widely studied in statistics and probability Karlin & Rinott (1980). In particular, isotropic Gaussian distributions are $\text{MTP}_2$.

**Proposition 9** (One-step DDIM). *With all the notations as above, fix $t > 0$, assume the distribution $p_t$ is MTP$_2$ with curvature bound $\kappa_t$, and*

$$a_t \ \geq \ c_t \kappa_t,$$

*then $F_t$ is coordinatewise nondecreasing: $\partial_{x_j} F_{t,i}(x) \geq 0$ for all $i, j$ and all $x$. Moreover, for any linear statistics $S$, we have $\mathrm{Corr}(S \circ F_t(Z)), S \circ F_t(-Z)) \leq 0$.*

*Proof.* Differentiate componentwise:

$$\frac{\partial F_{t,i}}{\partial x_j}(x) \ = \ a_t \, \delta_{ij} \ + \ c_t \, \partial_{ij}^2 \log p_t(x).$$

For $i \neq j$ this is $\geq 0$ by $c_t \geq 0$ and the mixed second derivative condition. For $i = j$,

$$\frac{\partial F_{t,i}}{\partial x_i}(x) \ \geq \ a_t + c_t \left( -\kappa_t \right) \ \geq \ 0.$$

Let $G = S \circ F_t$. Since $F_t$ is coordinately non-decreasing and $S$ is linear, we have that $G = S \circ F_t$ is partially monotone. Thus $\mathrm{Corr}(S \circ F_t(Z)), S \circ F_t(-Z)) \leq 0$ by the multivariate FKG inequality and Proposition 8. □

The next result shows that the idealized DDIM trajectory continues to generate negative correlation, as long as the curvature bound holds uniformly over all steps.

**Proposition 10** (Global DDIM chain). *With all the notations as above. Assume the distribution $p_t$ is MTP$_2$ with curvature bound $\kappa_t$, and*

$$a_t \ \geq \ c_t \kappa_t$$

*for every $t$. Then $\mathrm{DDIM_I} = F_1 \circ F_2 \circ F_3 \circ \cdots \circ F_T$ is coordinatewise nondecreasing. Moreover, for any linear statistics $S$, we have $\mathrm{Corr}(S \circ \mathrm{DDIM_I}(Z)), S \circ \mathrm{DDIM_I}(-Z)) \leq 0$.*

*Proof.* Since each $F_t$ is coordinatewise nondecreasing, their composition $F_1 \circ \cdots \circ F_T$ is also coordinatewise nondecreasing. Therefore, the composition $S \circ \mathrm{DDIM_I}$ is partially monotone, thus $\mathrm{Corr}(S \circ \mathrm{DDIM_I}(Z)), S \circ \mathrm{DDIM_I}(-Z)) \leq 0$ as claimed. □

**Remark 1.** *All these results remain valid if we replace the exact score $\nabla \log p_t$ by the neural network $s_\theta$, up to a constant rescaling.*

**Remark 2.** *We note that the condition $a_t \geq c_t \kappa_t$ is satisfied when $\kappa_t = O(1)$ and the probability flow ODE is discretized with sufficiently small step size. For the deterministic DDIM update under a variance-preserving schedule,*

$$a_t = \sqrt{\frac{\alpha_{t-1}}{\alpha_t}}, \qquad c_t = \sqrt{1 - \alpha_{t-1}} - \sqrt{\frac{\alpha_{t-1}}{\alpha_t}} \sqrt{1 - \alpha_t}.$$

*Thus $a_t$ is of unit order, and $c_t$ captures the difference $|\alpha_{t-1} - \alpha_t|$. In the continuous-time limit with step size $\Delta t$ and a smooth function $\alpha(t)$, a Taylor expansion yields $a_t = 1 + O(\Delta t)$ and $c_t = O(\Delta t)$; thus $a_t \geq c_t \kappa_t$ holds when $\Delta t$ is sufficiently small and $\kappa_t$ stays bounded.*

## C  ADDITIONAL EXPERIMENTS

### C.1  CORRELATION EXPERIMENT SETUP ON OTHER MODELS

#### C.1.1  CONSISTENCY MODEL

We study both unconditional and conditional consistency models using publicly available checkpoints provided in Hugging Face's Diffusers library (von Platen et al., 2022): openai/diffusers-cd_imagenet64_l2, openai/diffusers-cd_cat256_l2, and openai/diffusers-cd_bedroom256_l2.

For unconditional models, we evaluate pre-trained models on LSUN-Cat and LSUN-Bedrooom (Yu et al., 2015). For each dataset, 1,600 image pairs are generated under both PN and RR noise sampling with 1 DDIM steps.

For conditional models, we evaluate pre-trained models on ImageNet on 32 classes (Deng et al., 2009). For each class, 100 PN and RR pairs are generated with 1 steps.

### C.1.2 GENERATIVE MODELS BEYOND DIFFUSION

A mentioned in Section 3, we evaluate correlation on a VAE and flow-based models. Unlike the diffusion models, which we use public pre-trained checkpoints, both of these models required explicit training before evaluation. Here we describe the training setup and generation procedure.

**VAE:** We train the unconditional VAE on MNIST following the publicly available implementation provided in Francis (2022). The VAE consists of a simple convolutional encoder–decoder architecture with a Gaussian latent prior. After training, we generate 1,600 paired samples under PN and RR schemes, respectively.

**Glow:** We train the class-conditional Glow model (Kingma & Dhariwal, 2018) on CIFAR-10 using the normflows library. The architecture follows a multiscale normalizing flow design and Glow blocks. After training, we generated 100 PN and 100 RR pairs per class.

### C.2 IMAGE EDITING

We apply our antithetic initial noise strategy to the image editing algorithm FlowEdit (Kulikov et al., 2024), which edits images toward a target text prompt using pre-trained flow models.

In Algorithm 1 of FlowEdit, at each timestep, the algorithm samples $n_{\text{avg}}$ random noises $Z \sim \mathbb{N}(0, 1)$ to create noisy versions of the source image, computes velocity differences between source and target conditions, and averages these directions to drive the editing process.

In the $n_{\text{avg}} = 2$ setting, we replace the two independent random samples with antithetic noises: for each $Z \sim \mathbb{N}(0, I)$ we also use $-Z$ and average the two velocity updates.

We compare on 76 prompts provided in FlowEdit's official GitHub repository. For each prompt, we generate 10 images using both PN and RR. All other parameters follow the repository defaults. We evaluate performance using CLIP(semantic text–image alignment; higher is better) and LPIPS (perceptual distance to the source; lower is better), which jointly measure text adherence and structure preservation.

As a result, PN improves the mean CLIP score, winning in 56.59% of all pairwise comparisons. It also reduces LPIPS, winning in 81.58% of all pairwise comparisons.

### C.3 IMPLEMENTATION DETAIL

We use pretrained models from Hugging Face's Diffusers library (von Platen et al., 2022): google/ddpm-church-256, google/ddpm-cifar10-32, and google/ddpm-celebahq-256 for unconditional diffusion; Stable Diffusion v1.5 for text-to-image; and the original repository from (Chung et al., 2023a) for guided generation. Experiments were run on eight NVIDIA L40 GPUs. The most intensive setup—Stable Diffusion—takes about five minutes to generate 100 images for a single prompt.

### C.4 ADDITIONAL EXPERIMENTS ON PIXEL-WISE SIMILARITY

### C.4.1 DDPM

**Antithetic sampling setup:** Unlike DDIM, where the sampling trajectory is deterministic once the initial noise is fixed, DDPM adds a random Gaussian noise at every timestep. The update function in DDPM is

$$x_{t-1} = \frac{1}{\sqrt{\alpha_t}}(x_t - \frac{1 - \alpha_t}{\sqrt{1 - \alpha_t}}\epsilon_\theta(x_t, t)) + \sigma_t z_t$$

and $z_t \sim \mathbb{N}(0, 1)$ if $t > 1$, else $z = 0$. Therefore, in DDPM, antithetic sampling requires not only negating the initial noise but also negating *every noise* $z_t$ added at each iteration.

Table 5 and Table 6 report the standard Pearson correlations and centralized correlation between pixel values of image pairs produced using the same pre-trained models under PN and RR noise schemes using DDPM (default 1000 steps) across CIFAR10, CelebA-HQ, and LSUN-Church. For each dataset, we follow the same setup explained in Section 3. We calculate the pixelwise correction with 1600 antithetic noise pairs and 1600 independent noises.

Consistent with the behavior observed in DDIM, PN pairs in DDPM samples also exhibit negative correlation. Using the standard Pearson correlation, the mean correlation for PN pairs is strongly negative in all datasets—CIFAR10 ($-0.73$), Church ($-0.45$), and in the more identity-consistent CelebA ($-0.18$).

The centralized correlation analysis further sharpens this contrast: mean PN correlations are substantially lower (CIFAR10 $-0.80$, CelebA $-0.67$, Church $-0.65$). These results confirm again that PN noise pairs consistently introduce strong negative dependence, while RR pairs remain close to uncorrelated or weakly positive.

Table 5: DDPM standard Pearson correlation coefficients for PN and RR pairs

| Statistic | CIFAR10 | | CelebA | | Church | |
|---|---|---|---|---|---|---|
| | PN | RR | PN | RR | PN | RR |
| Mean | -0.73 | 0.04 | -0.18 | 0.29 | -0.45 | 0.11 |
| Min | -0.95 | -0.61 | -0.80 | -0.50 | -0.90 | -0.60 |
| 25th percentile | -0.81 | -0.12 | -0.31 | 0.16 | -0.55 | -0.02 |
| 75th percentile | -0.66 | 0.20 | -0.05 | 0.44 | -0.36 | 0.23 |
| Max | -0.21 | 0.80 | 0.44 | 0.78 | 0.07 | 0.80 |

Table 6: DDPM centralized Pearson correlation coefficients for PN and RR pairs

| Statistic | CIFAR10 | | CelebA | | Church | |
|---|---|---|---|---|---|---|
| | PN | RR | PN | RR | PN | RR |
| Mean | -0.80 | 0.01 | -0.67 | -0.00 | -0.65 | -0.00 |
| Min | -0.96 | -0.61 | -0.89 | -0.60 | -0.93 | -0.56 |
| 25th percentile | -0.87 | -0.15 | -0.73 | -0.15 | -0.72 | -0.12 |
| 75th percentile | -0.75 | 0.16 | -0.62 | 0.15 | -0.60 | 0.11 |
| Max | -0.23 | 0.76 | 0.14 | 0.64 | -0.32 | 0.72 |

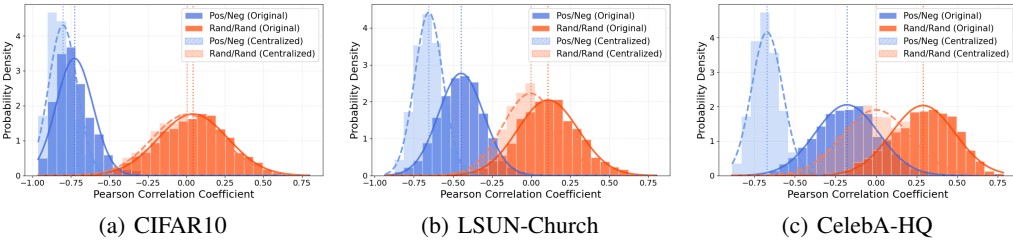

| (a) CIFAR10 | (b) LSUN-Church | (c) CelebA-HQ |
|---|---|---|

Figure 5: Pearson Correlation histograms for PN and RR pairs across three datasets using DDPM. Dashed lines indicate the mean Pearson correlation for each group.

### C.4.2 DIFFUSION POSTERIOR SAMPLING (DPS)

The same pattern persists in the setting of using the diffusion model as a prior for posterior sampling, too, which has been utilized to solve various inverse problems, such as inpainting, super-resolution, and Gaussian deblurring.

Since there is a ground truth image available in the image restoration task, the standard pixel-wise correlation is calculated using the difference between reconstructed images and the corresponding ground truth, and the centralized correlation is calculated using the same definition described in 3. Although the overall standard correlation values are shifted up, due to the deterministic conditioning, the posterior nature of sampling—PN pairs still shows significantly lower correlations than RR pairs across all tasks.

For the standard Pearson correlation, the mean PN correlations range from $0.20$ to $0.27$, while RR correlations consistently lie above $0.50$. For the centralized correlation, PN correlations are strongly negative across all tasks (means around $-0.72$). In contrast, RR pairs remain centered near zero (mean correlations around $-0.01$ to $-0.02$).

Table 7: DPS Pearson correlation coefficients for PN and RR pairs

| Statistic | Inpainting | | Gaussian Deblur | | Super-resolution | |
| --- | --- | --- | --- | --- | --- | --- |
| | PN | RR | PN | RR | PN | RR |
| **Mean** | 0.28 | 0.57 | 0.27 | 0.57 | 0.20 | 0.53 |
| **Min** | -0.14 | 0.05 | -0.22 | 0.13 | -0.34 | 0.01 |
| **25th percentile** | 0.19 | 0.52 | 0.17 | 0.51 | 0.10 | 0.47 |
| **75th percentile** | 0.36 | 0.63 | 0.36 | 0.63 | 0.30 | 0.60 |
| **Max** | 0.66 | 0.83 | 0.62 | 0.83 | 0.64 | 0.81 |

Table 8: DPS centralized correlation coefficients for PN and RR pairs

| Statistic | Inpainting | | Gaussian Blur | | Super-resolution | |
| --- | --- | --- | --- | --- | --- | --- |
| | PN | RR | PN | RR | PN | RR |
| **Mean** | -0.72 | -0.02 | -0.71 | -0.01 | -0.72 | -0.01 |
| **Min** | -0.86 | -0.43 | -0.84 | -0.36 | -0.87 | -0.35 |
| **25th percentile** | -0.76 | -0.07 | -0.75 | -0.07 | -0.76 | -0.08 |
| **75th percentile** | -0.69 | 0.04 | -0.67 | 0.05 | -0.69 | 0.05 |
| **Max** | -0.14 | 0.47 | 0.01 | 0.48 | -0.14 | 0.43 |

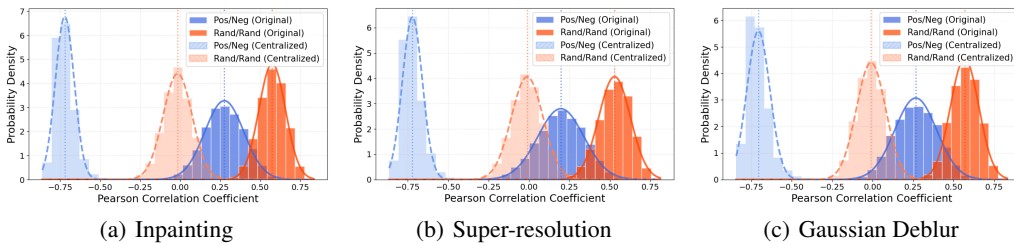

(a) Inpainting        (b) Super-resolution        (c) Gaussian Deblur

Figure 6: Pearson Correlation histograms for PN and RR pairs across three tasks in DPS. Dashed lines indicate the mean Pearson correlation for each group.

### C.4.3 WASSERSTEIN DISTANCE

To complement the correlation-based analyses in the main text, we also evaluate similarity using the Wasserstein distance, a measure of distributional discrepancy. It quantifies the minimal "effort" required to transform one probability distribution into another, which means lower Wasserstein values indicate closer alignment between the two distributions, while higher values indicate larger differences.

To calculate Wasserstein distances, we treat each generated image pair as a sample from a distribution under the sampling scheme, PN or RR. As shown in Table 9, PN consistently exhibits larger Wasserstein distances than RR across nearly all models, and the differences are statistically significant

at all $p < 10^{-10}$. This implies that antithetic initial noises lead to more divergent distributions than random sampling and confirms our results from the correlation analysis.

Table 9: Wasserstein Distance, shown are means (SD) with corresponding t-statistics and p-values.

| Model | Wasserstein Distance | | t-stats (p) |
| --- | --- | --- | --- |
| | PN | RR | |
| LSUN-Church | 0.16 (0.10) | 0.12 (0.07) | $t = 12.09,\ p = 0$ |
| CIFAR-10 | 0.19 (0.14) | 0.15 (0.09) | $t = 8.30,\ p = 0$ |
| CelebA-HQ | 0.17 (0.11) | 0.12 (0.07) | $t = 14.50,\ p = 0$ |
| SD 1.5 | 0.10 (0.06) | 0.09 (0.04) | $t = 33.11,\ p = 0$ |
| DiT | 0.19 (0.15) | 0.14 (0.12) | $t = 14.17,\ p = 0$ |
| VAE | 0.07 (0.05) | 0.05 (0.03) | $t = 13.24,\ p = 0$ |
| Glow | 0.15 (0.09) | 0.12 (0.08) | $t = 7.92,\ p = 0$ |

### C.4.4 INFLUENCE OF THE CFG SCALE ON CORRELATION

We extend our pixel-correlation analysis on conditional diffusion models across various Classifier-Free Guidance (CFG) scales. CFG (Ho & Salimans, 2022) is a technique that strengthens conditioning in diffusion models by interpolating between conditional and unconditional score estimates. The guidance scale controls this interpolation strength, with higher values producing samples more aligned with the conditioning signal such as prompt or class.

We conducted experiments on both Stable Diffusion 1.5 (SD1.5) and DiT using CFG scales {1, 3, 5, 7, 9}. For each setting, we generated 100 PN and RR noise pairs across 25 prompts/classes, measuring both standard and centralized correlations between the image samples.

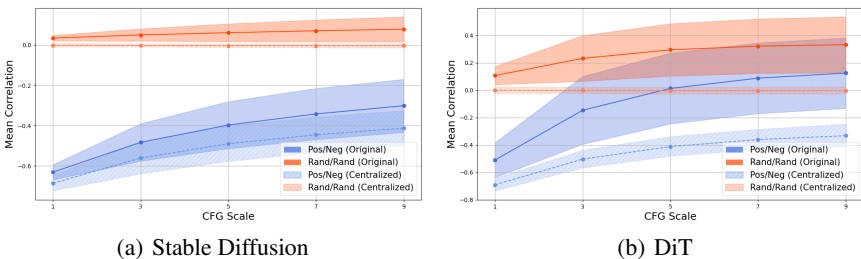

(a) Stable Diffusion  (b) DiT

Figure 7: Average pixel correlation versus CFG scale for SD1.5 and DiT. Both standard and centralized correlations are shown. Shaded regions indicate standard deviation of mean correlation across 25 prompts/classes

As show in Figure 7, for all CFG scales, negated noise continues to produce strongly negatively correlated samples. At the same time, the standard correlation for both PN and RR grows as the CFG scale increases. This can be explained as follows: larger CFG values pull samples more strongly toward the conditioning signal (prompt/class), effectively shrinking the space of plausible outputs. As samples concentrate more tightly around the target distribution, they become more similar to one another. Thus, the correlations of both PN and RR pairs increase, while PN remains much more negative than RR.

### C.4.5 PARTIAL NEGATION OF NOISE VECTORS

We conducted experiments to evaluate how localized antithetic noise influences outputs of unconditional diffusion models trained on LSUN-Church and LSUN-Bedroom. For each dataset, we generated 200 image pairs by sampling $Z_1 \sim \mathbb{N}(0, I)$ and constructing $Z_2$ by negating the upper half of $Z_1$ while leaving the lower half unchanged.

We calculate the Pearson standard correlation and the centered correlation between corresponding halves to quantify spatial correspondence induced by the antithetic manipulation. As shown in

Figure 8, the top halves exhibit sharply negative correlations, and the bottom halves are highly positively correlated and visually almost identical (Figure 27, 28). This shows that the negative correlation effect acts locally in noise space and directly affects the corresponding spatial regions in the generated images.

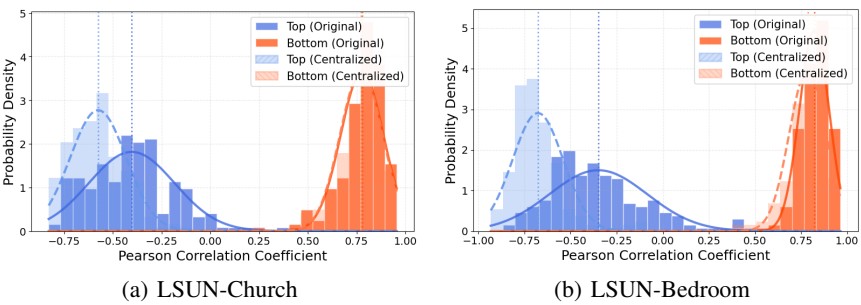

(a) LSUN-Church                                (b) LSUN-Bedroom

Figure 8: Average pixel correlation of top-half and bottom-half across generated image pairs.

## C.5 ADDITIONAL EXPERIMENTS ON THE SYMMETRY CONJECTURE

We run additional experiments to validate the conjecture in Section 3.2. Using a pretrained score network and a 50-step DDIM sampler, we evaluated both CIFAR-10 and Church. For each dataset, we selected five random coordinates in the $C \times H \times W$ tensor (channel, height, width). At every chosen coordinate we examined the network output at time steps $t = 1, \ldots, 20$ (small $t$ is close to the final image, large $t$ is close to pure noise). For each $t$ we drew a standard Gaussian sample $\mathbf{x} \sim \mathbb{N}(0, I_d)$ and computed the value of $\epsilon_\theta^{(t)}(c\,\mathbf{x})$ at the selected coordinate. The resulting plots appear in Figures 9–18.

To measure how much a one–dimensional function resembles an antisymmetric shape, we introduce the *affine antisymmetry score*

$$\text{AS}(f) := 1 - \frac{\int_{-1}^{1}(0.5f(-x) + 0.5f(x) - \bar{f})^2}{\int_{-1}^{1}(f(x) - \bar{f})^2}$$

where $\bar{f} := \int_{-1}^{1} f \, dx / 2$ is the average value of $f$ on $[-1, 1]$.

The integral's numerator is the squared average distance between the antithetic mean $0.5\,f(-x) + 0.5\,f(x)$ and the overall mean $\bar{f}$, while the denominator is the full variance of $f$ over the interval.

The antisymmetry score is well-defined for every non-constant function $f$; it takes values in the range $[0, 1]$ and represents the fraction of the original variance that is eliminated by antithetic averaging. The score attains $\text{AS}(f) = 1$ exactly when the antithetic sum $f(x) + f(-x)$ is constant, that is, when $f$ is perfectly *affine-antisymmetric*. Conversely, $\text{AS}(f) = 0$ if and only if $f(-x) = f(x) + c$ for some constant $c$, meaning $f$ is *affine-symmetric*.

For each dataset we have $100$ ($5$ coordinates $\times 20$ time steps) scalar functions; the summary statistics of their AS scores are listed in Table 10. Both datasets have very high AS scores: the means exceed $0.99$ and the $10\%$ quantiles are above $0.97$, indicating that antithetic pairing eliminates nearly all variance in most cases. Even the lowest scores (about $0.77$) still remove more than three-quarters of the variance.

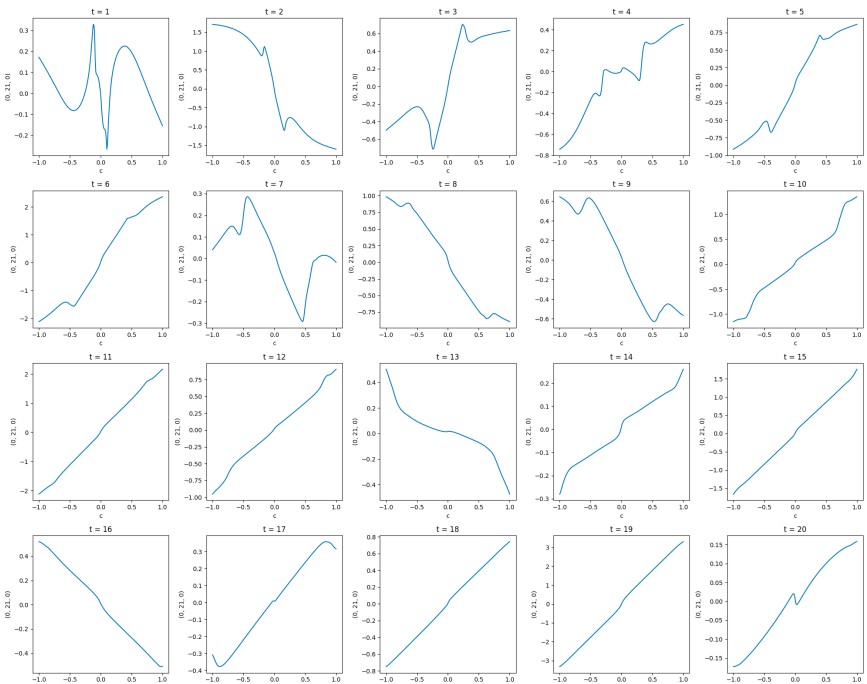

Figure 9: CIFAR10: Coordinate $(0, 21, 0)$

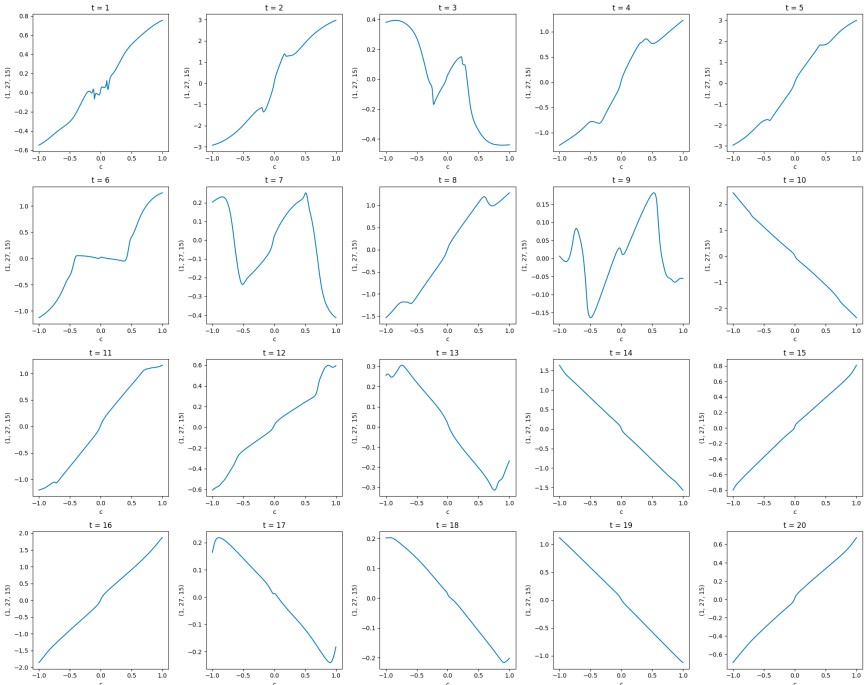

Figure 10: CIFAR10: Coordinate $(1, 27, 15)$

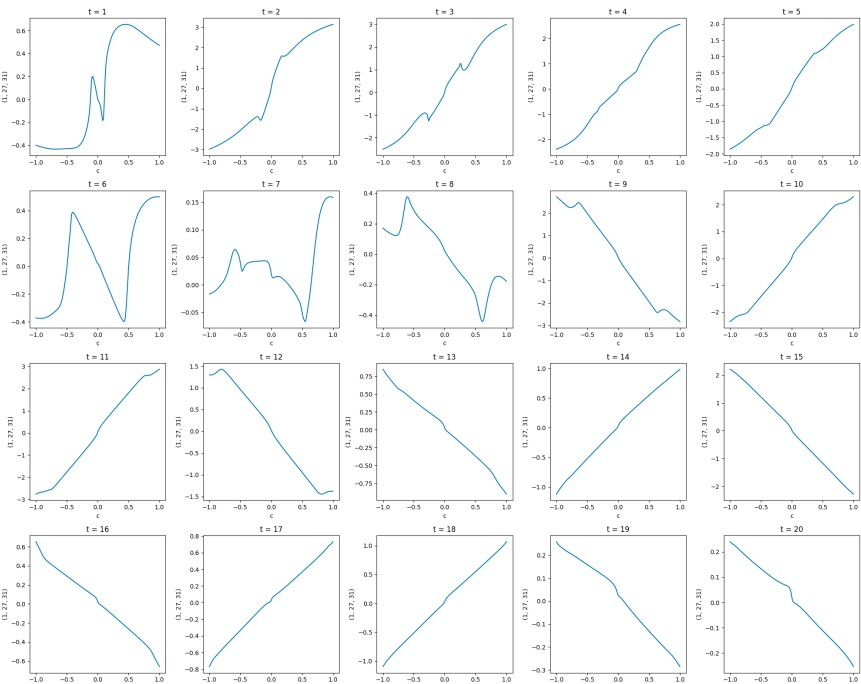

Figure 11: CIFAR10: Coordinate $(1, 27, 31)$

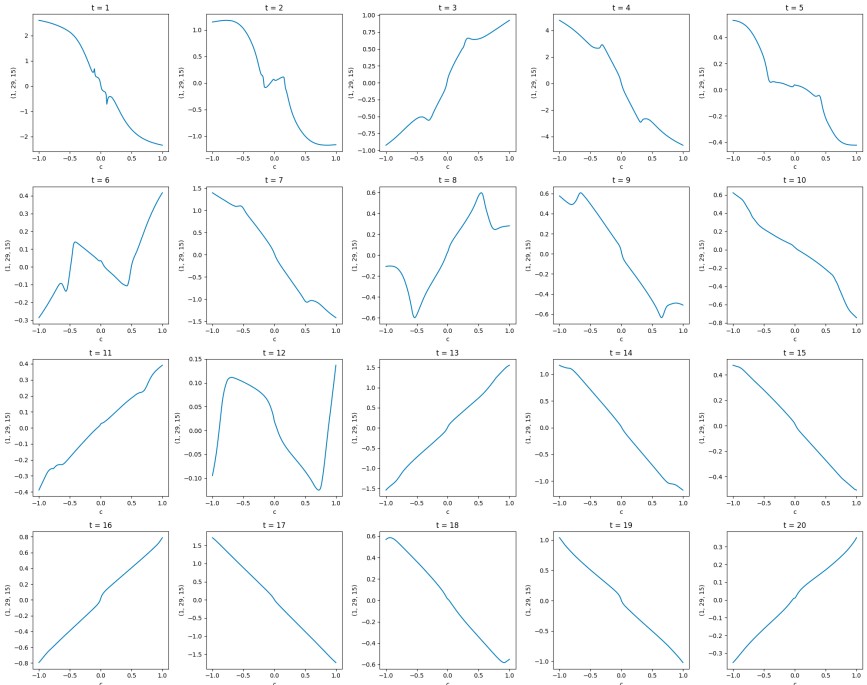

Figure 12: CIFAR10: Coordinate $(1, 29, 15)$

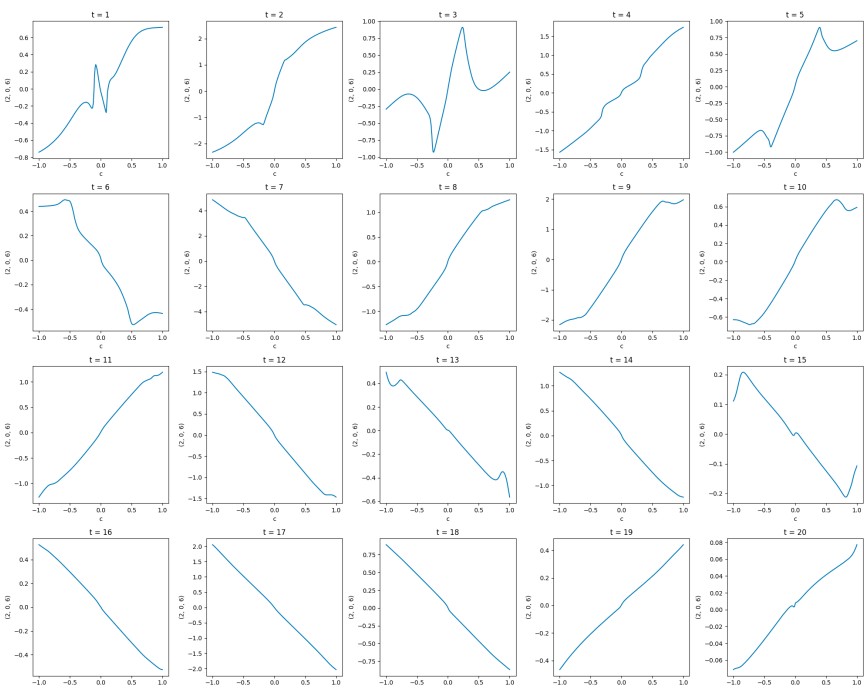

Figure 13: CIFAR10: Coordinate $(2, 0, 6)$

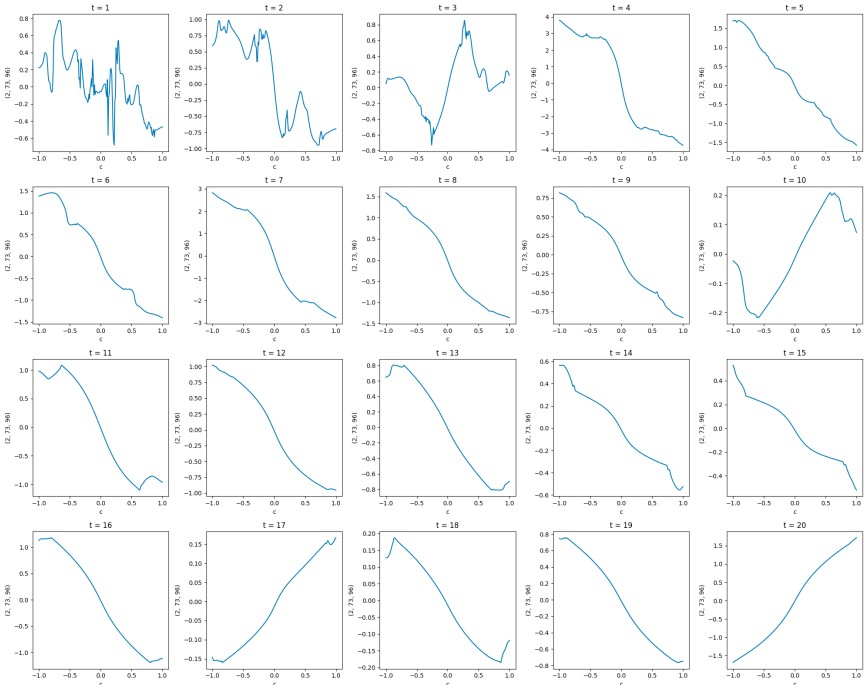

Figure 14: Church: Coordinate $(2, 73, 76)$

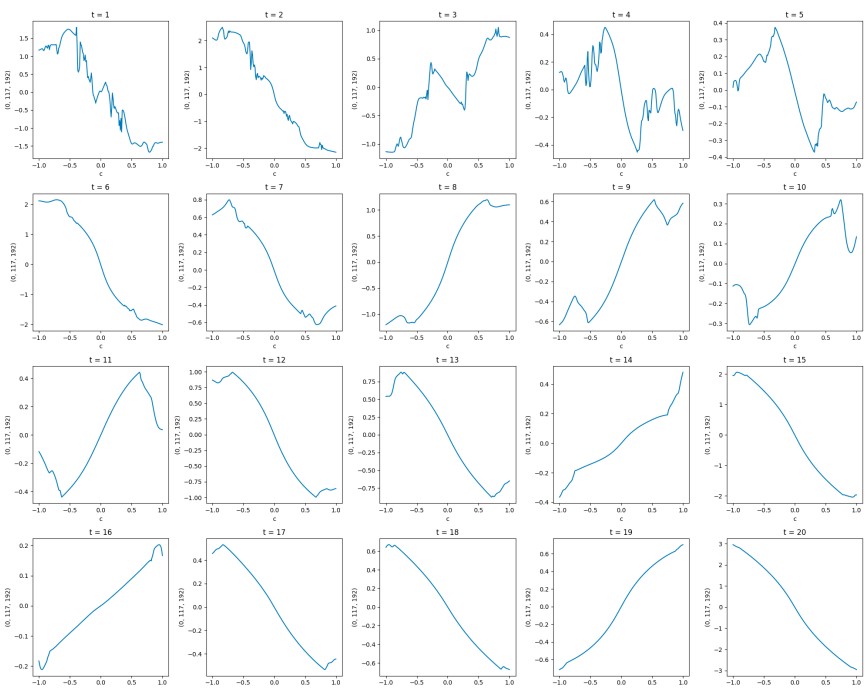

Figure 15: CIFAR10: Coordinate $(0, 117, 192)$

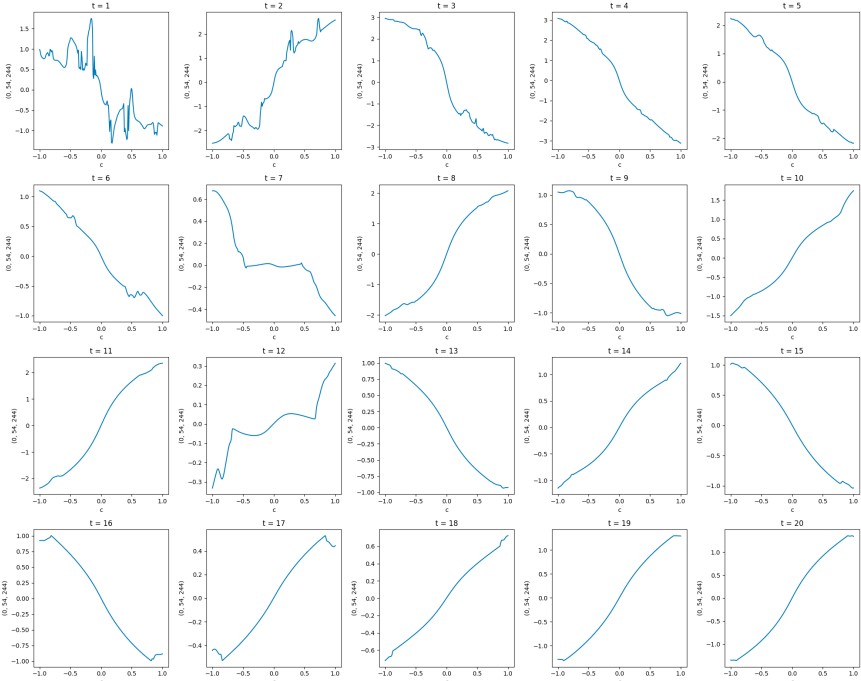

Figure 16: Church: Coordinate $(0, 54, 244)$

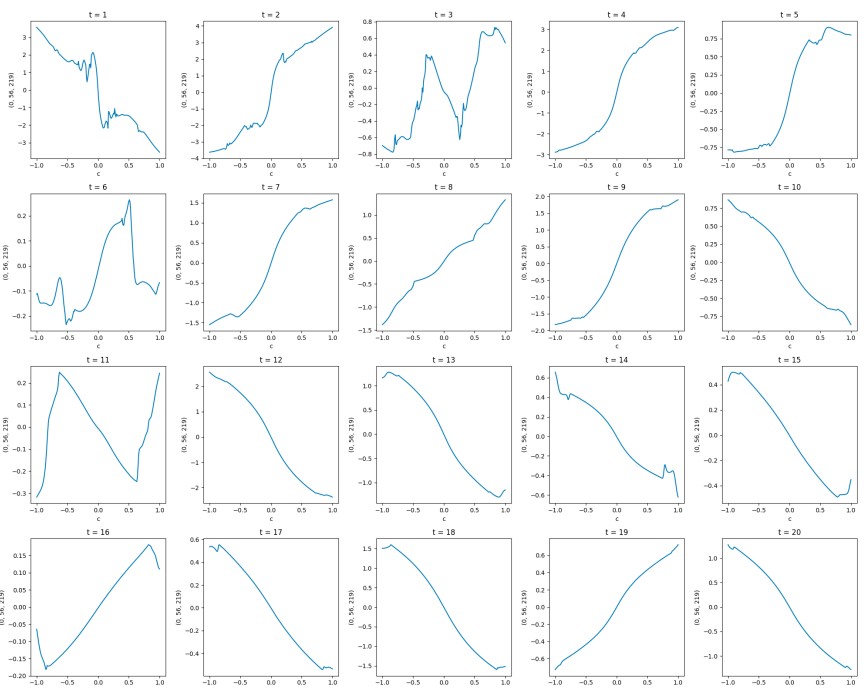

Figure 17: CIFAR10: Coordinate $(0, 56, 219)$

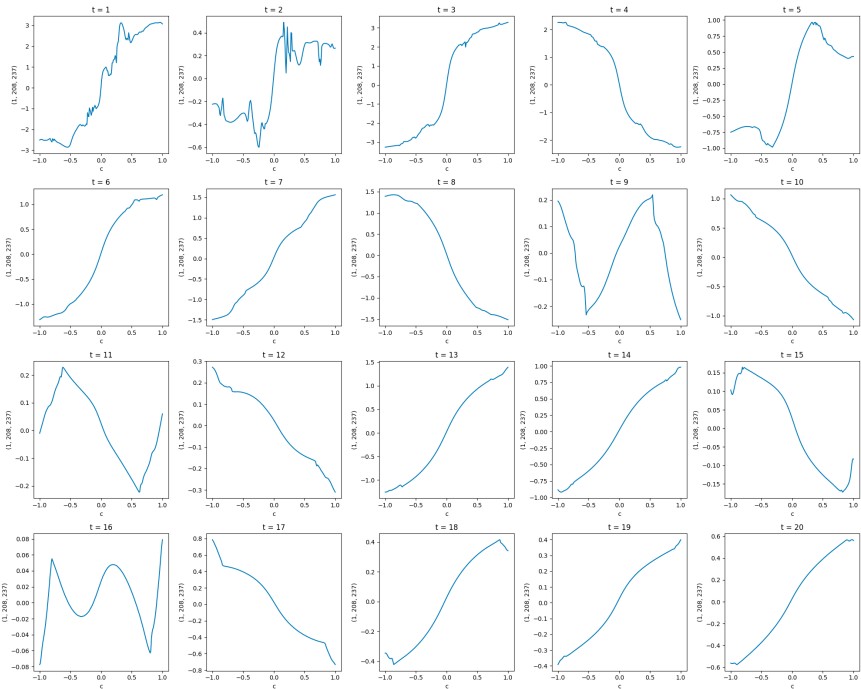

Figure 18: Church: Coordinate $(1, 208, 237)$

Table 10: Affine antisymmetry score

| Dataset | CIFAR10 | Church |
|---|---|---|
| **Mean** | 0.9932 | 0.9909 |
| Min | 0.7733 | 0.7710 |
| 1% quantile | 0.9104 | 0.8673 |
| 2% quantile | 0.9444 | 0.8768 |
| 5% quantile | 0.9690 | 0.9624 |
| 10% quantile | 0.9865 | 0.9750 |
| Median | 0.9992 | 0.9995 |

## C.6   ADDITIONAL EXPERIMENTS FOR UNCERTAINTY QUANTIFICATION

### C.6.1   UNCERTAINTY QUANTIFICATION

The image metrics used in uncertainty quantification are used to capture different aspects of pixel intensity, color distribution, and perceived brightness.

**Mean pixel value** is defined as the average of all pixel intensities across the image (including all channels). Formally, for an image $I \in \mathbb{R}^{C \times H \times W}$, the mean is computed as

$$\mu = \frac{1}{HWC} \sum_{i=1}^{C} \sum_{j=1}^{H} \sum_{k=1}^{W} I_{i,j,k}.$$

**Brightness** is calculated using the standard CIE formula: $0.299 \cdot R + 0.587 \cdot G + 0.114 \cdot B$ to produce a grayscale value at *each pixel*, where $R, G, B$ is the red, green, and blue color value of the given pixel. It is widely used in video and image compression, and approximates human visual sensitivity to color. The brightness of an image is then the average of the grayscale value across all pixels.

**Contrast** is computed as the difference in average pixel intensity between the top and bottom halves of an image. Let $x \in [0,1]^{C \times H \times W}$ be the normalized image, we define contrast as $100 \cdot (\mu_{\text{top}} - \mu_{\text{bottom}})$, where $\mu_{\text{top}}$ and $\mu_{\text{bottom}}$ are the average intensities over the top and bottom halves, respectively.

**Centroid** measures the coordinate of the brightness-weighted center of mass of the image. For scalar comparison, we focus on the vertical component of the centroid to assess spatial uncertainty. After converting to grayscale $M \in \mathbb{R}^{H \times W}$ by averaging across channels, we treat the image as a 2D intensity distribution and compute the vertical centroid as

$$\frac{1}{\sum_{i=1}^{H} \sum_{j=1}^{W} M_{i,j}} \sum_{i=1}^{H} \sum_{j=1}^{W} i \cdot M_{i,j},$$

where $i$ denotes the row index.

### C.6.2   QMC EXPERIMENTS

**RQMC confidence interval construction:**   For the RQMC experiments, we consider $R$ independent randomization of a Sobol' point set of size $n$, with a fixed budget of $N = Rn$ function evaluations. Denote the RQMC point set in the $r$-th replicate as $\{\mathbf{u}_{r,k}\}_{1 \le k \le n} \subset [0,1]^d$, where $\mathbf{u}_{r,k} \sim \text{Unif}([0,1]^d)$. Applying the Gaussian inverse cumulative distribution function $\Phi^{-1}$ to each coordinate of $\mathbf{u}_{r,k}$ transforms the uniform samples to standard normal samples. Consequently, the estimate in each replicate is given by

$$\hat{\mu}_r^{\text{QMC}} = \frac{1}{n} \sum_{k=1}^{n} S\big(\text{DM}\big(\Phi^{-1}(\mathbf{u}_{r,k})\big)\big), \quad r = 1, \dots, R.$$

The overall point estimate is their average

$$\hat{\mu}_N^{\text{QMC}} = \frac{1}{R} \sum_{r=1}^{R} \hat{\mu}_r^{\text{QMC}}.$$

Let $(\hat{\sigma}_R^{\text{QMC}})^2$ be the sample variance of $\hat{\mu}_1^{\text{QMC}}, \ldots, \hat{\mu}_R^{\text{QMC}}$. The Student $t$ confidence interval is given by

$$\text{CI}_N^{\text{QMC}}(1-\alpha) = \hat{\mu}_N^{\text{QMC}} \pm t_{R-1,\,1-\alpha/2} \frac{\hat{\sigma}_R^{\text{QMC}}}{\sqrt{R}},$$

where $t_{R-1,\,1-\alpha/2}$ is the $(1-\alpha/2)$-quantile of the $t$-distribution with $R-1$ degrees of freedom. If the estimates $\hat{\mu}_r^{\text{QMC}}$ are normally distributed, this confidence interval has exact coverage probability $1-\alpha$. In general, the validity of Student $t$ confidence interval is justified by CLT. An extensive numerical study by L'Ecuyer et al. (2023) demonstrates that Student $t$ intervals achieve the desired coverage empirically.

**Exploring different configurations of $R$ and $n$** This experiment aims to understand how different configurations of $R$, the number of replicates, and $n$, the size of the QMC point set, affect the CI length of the RQMC method. The total budget of function evaluations is fixed at $Rn = 3200$, to be consistent with the AMC and MC experiments. We consider the four image metrics used in Section 4 and one additional image metric, MUSIQ.

For RQMC, each configuration was repeated five times, and the results were averaged to ensure stability. AMC and MC each consist of a single run over 3200 images. All experiments are conducted using the CIFAR10 dataset.

The results are shown in Table 11 and underlined values indicate the best CI length among the three QMC configurations, while bold values indicate the best CI length across all methods, including MC and AMC. For the brightness and pixel mean metrics, the configuration with point set size $n = 64$ and number of replicates $R = 50$ reduces CI length the most. In contrast and centroid, the configuration with larger $n$ has a better CI length. For MUSIQ, a more complex metric, changes in CI lengths across configurations are subtle, and the configuration with the largest $R$ has the shortest CI length. While no single configuration consistently advantages, all RQMC and antithetic sampling methods outperform plain MC.

Table 11: Average 95% CI length and relative efficiencies (vs MC baseline). The first three rows are for RQMC methods, with the configuration of $R \times n$ indicated in the first column.

| $R \times n$ | **Brightness** | | **Mean** | | **Contrast** | | **Centroid** | | **MUSIQ** | |
| | CI | Eff. | CI | Eff. | CI | Eff. | CI | Eff. | CI | Eff. |
|---|---|---|---|---|---|---|---|---|---|---|
| $25 \times 128$ | 0.37 | 29.81 | 0.40 | 25.71 | **0.22** | **24.57** | **0.04** | **8.17** | 0.13 | 1.11 |
| $50 \times 64$ | 0.35 | 32.05 | 0.39 | 26.94 | 0.22 | 23.43 | 0.04 | 7.35 | 0.13 | 1.13 |
| $200 \times 16$ | 0.47 | 18.38 | 0.49 | 17.30 | 0.29 | 14.20 | 0.05 | 5.84 | **0.12** | **1.21** |
| AMC | **0.35** | **32.66** | **0.39** | **27.12** | 0.23 | 22.05 | 0.04 | 6.96 | 0.13 | 1.06 |
| MC | 2.00 | - | 2.04 | - | 1.08 | - | 0.11 | - | 0.13 | - |

C.6.3 DPS EXPERIMENT IMPLEMENTATION

We evaluate the confidence interval length reduction benefits of antithetic initial noise across three common image inverse problems: super-resolution, Gaussian deblurring, and inpainting. For each task, the forward measurement operator is applied to the true image, and noisy observations are generated using Gaussian noise with $\sigma = 0.05$. We use the official implementation of DPS (Chung et al., 2023a) with the following parameters:

- **Super-resolution** uses the *super_resolution* operator with an input shape of $(1, 3, 256, 256)$ and an upsampling scale factor of 2. This models the standard bicubic downsampling scenario followed by Gaussian noise corruption.
- **Gaussian deblurring** employs the *gaussian_blur* operator, again with a kernel size of 61 but with intensity set to 1, which is intended to test the variance reduction in a simpler inverse scenario.

- **Inpainting** is set up using the *inpainting* operator with a random binary mask applied to the input image. The missing pixel ratio is drawn from a uniform range between 30% and 70%, and the image size is fixed at $256 \times 256$. A higher guidance scale (0.5) is used to compensate for the sparsity of observed pixels.

As shown in Table 3, AMC consistently achieves shorter CI lengths than MC across all tasks and metrics without sacrificing reconstruction quality, implied by the L1 and PSNR metrics to measure the difference between reconstructed images and ground truth images.

### C.6.4   DDS DATASET

Data used in the DDS experiment on uncertainty quantification (Section 4) were obtained from the NYU fastMRI Initiative database (Knoll et al., 2020; Zbontar et al., 2018). A listing of NYU fastMRI investigators, subject to updates, can be found at: fastmri.med.nyu.edu. The primary goal of fastMRI is to test whether machine learning can aid in the reconstruction of medical images.

# D    MORE VISUALIZATIONS

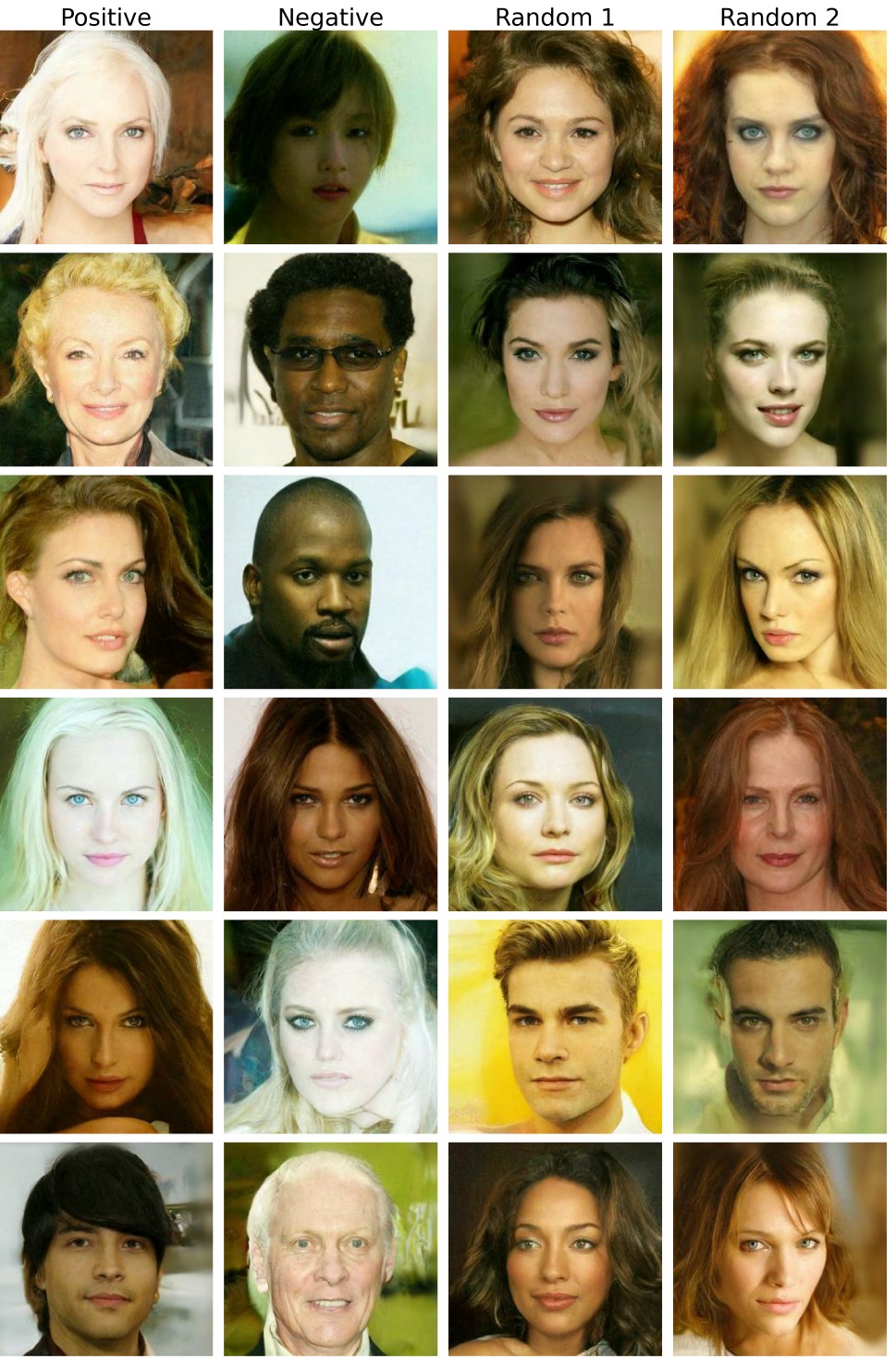

Figure 19: CelebA-HQ Image Generated

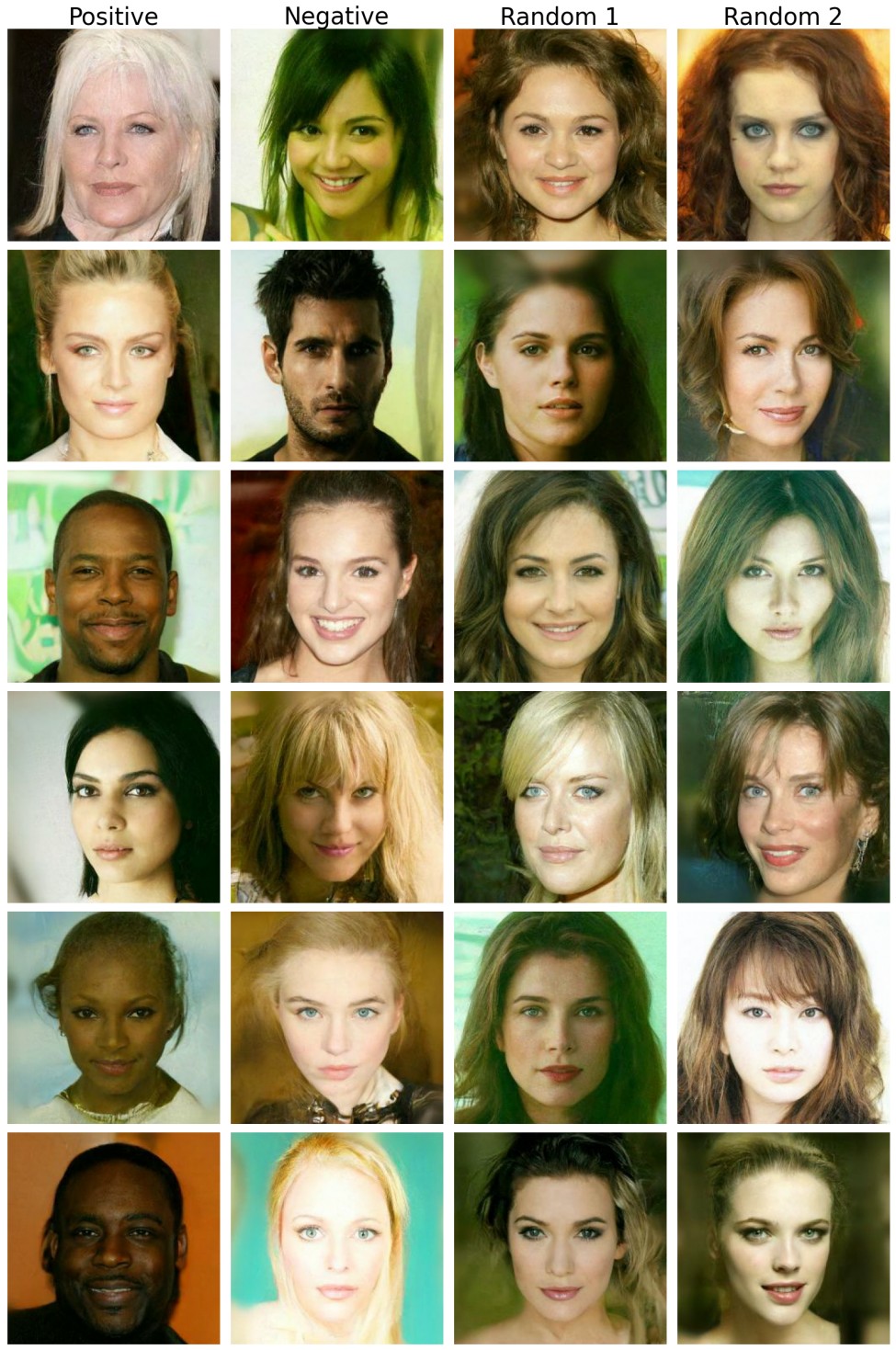

Figure 20: CelebA-HQ Image Generated

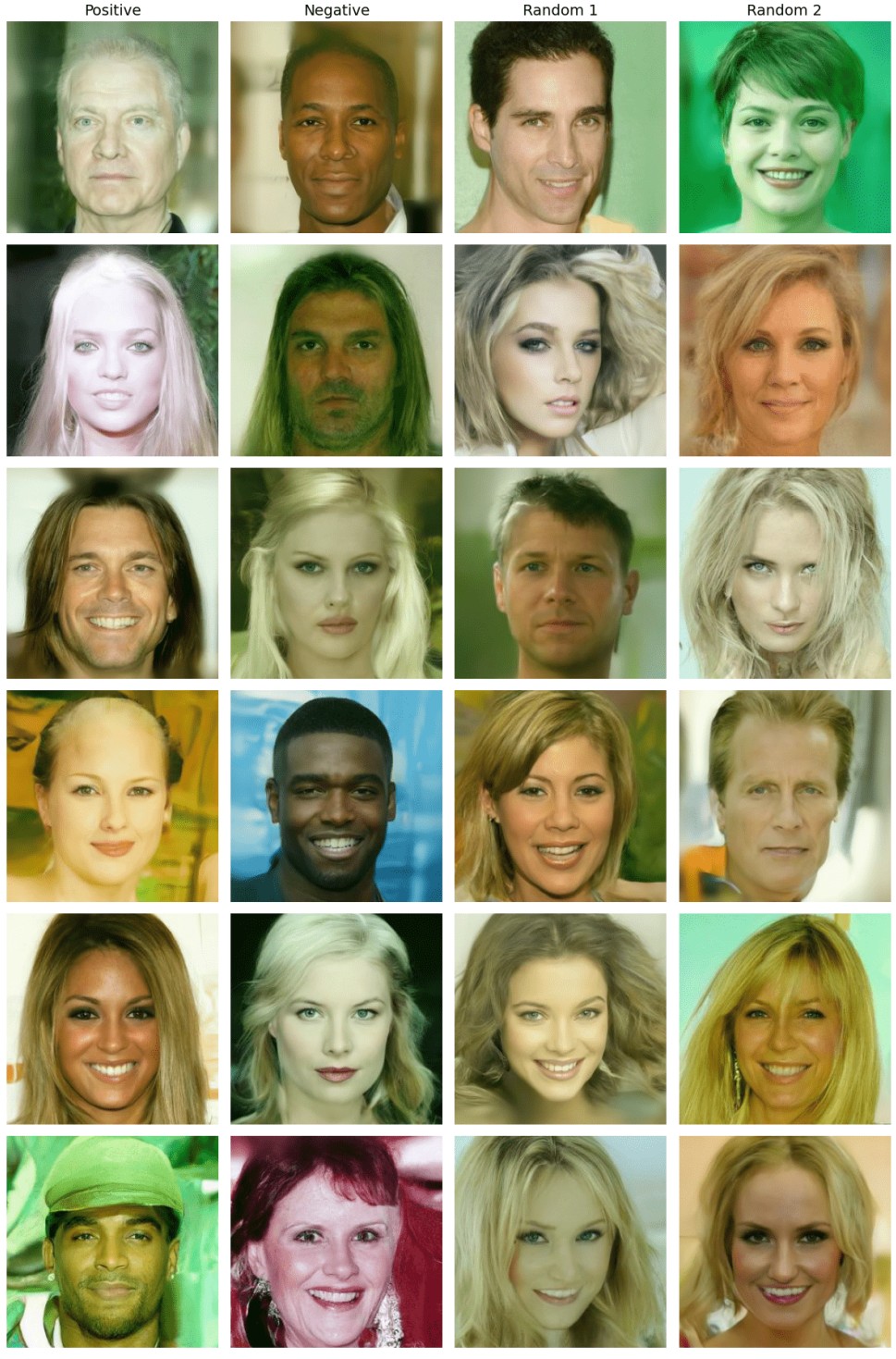

Figure 21: CelebA-HQ Image Generated

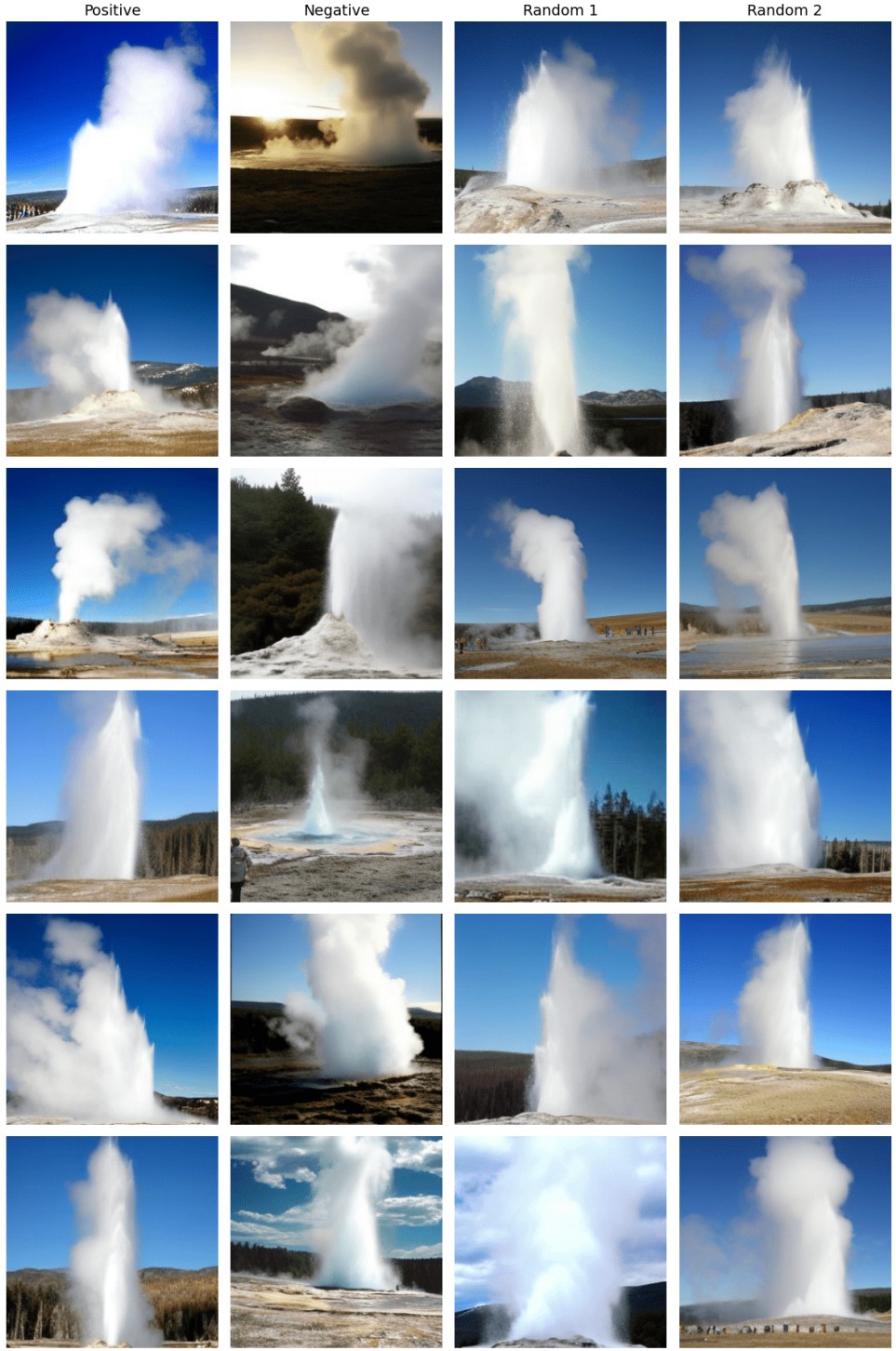

Figure 22: DiT Class 974: geyser

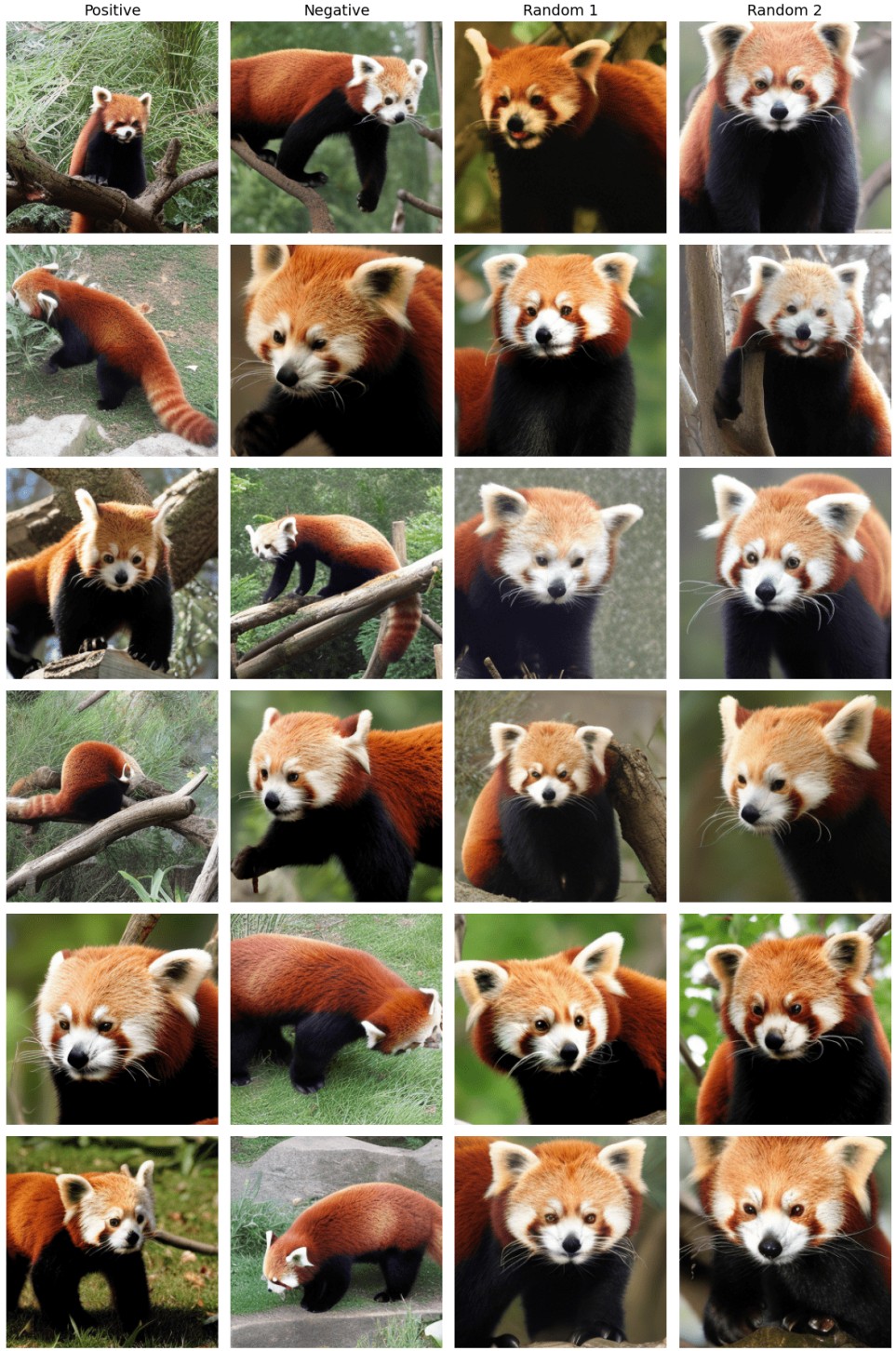

Figure 23: DiT Class 387: lesser panda, red panda

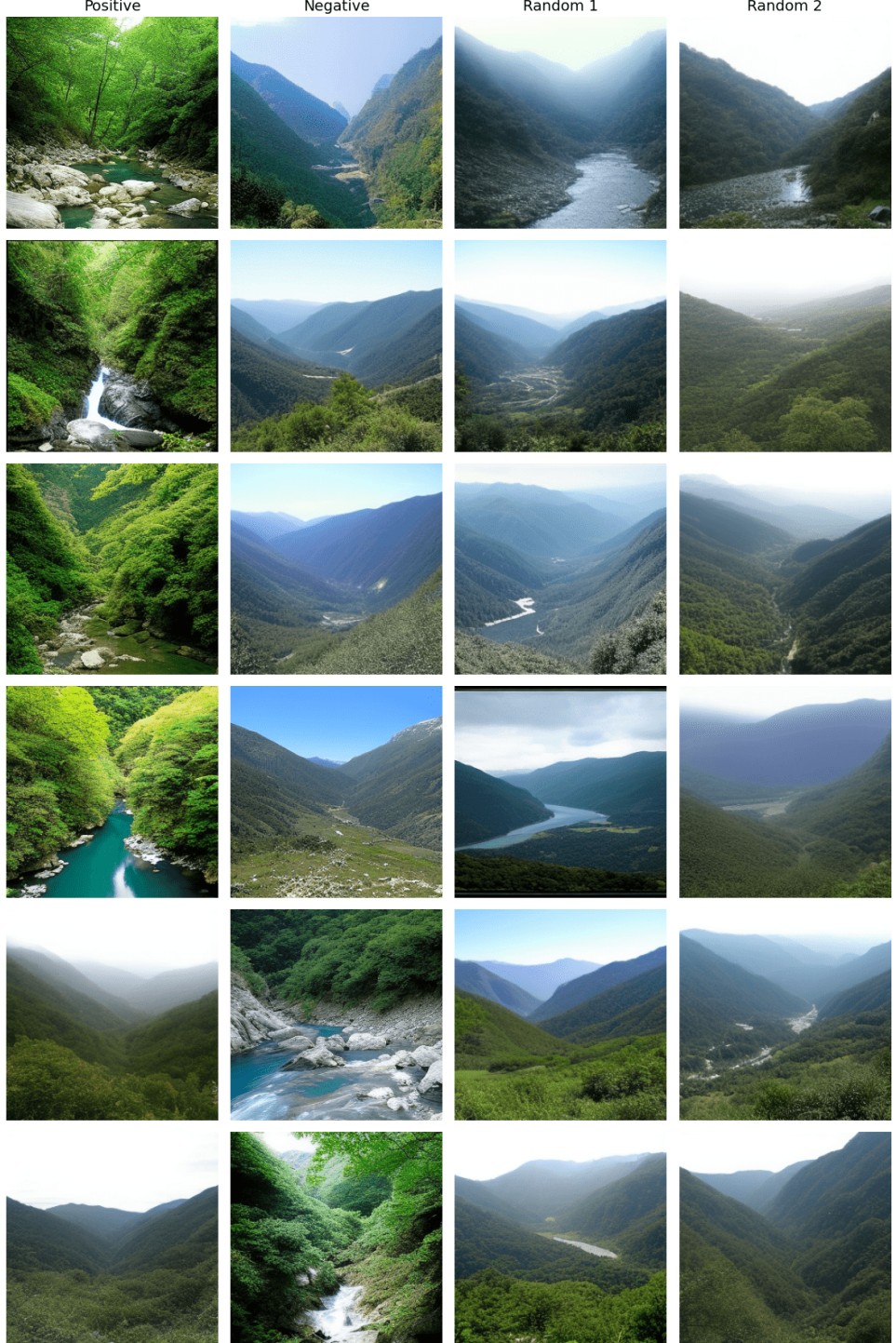

Figure 24: DiT Clas 979: valley, vale

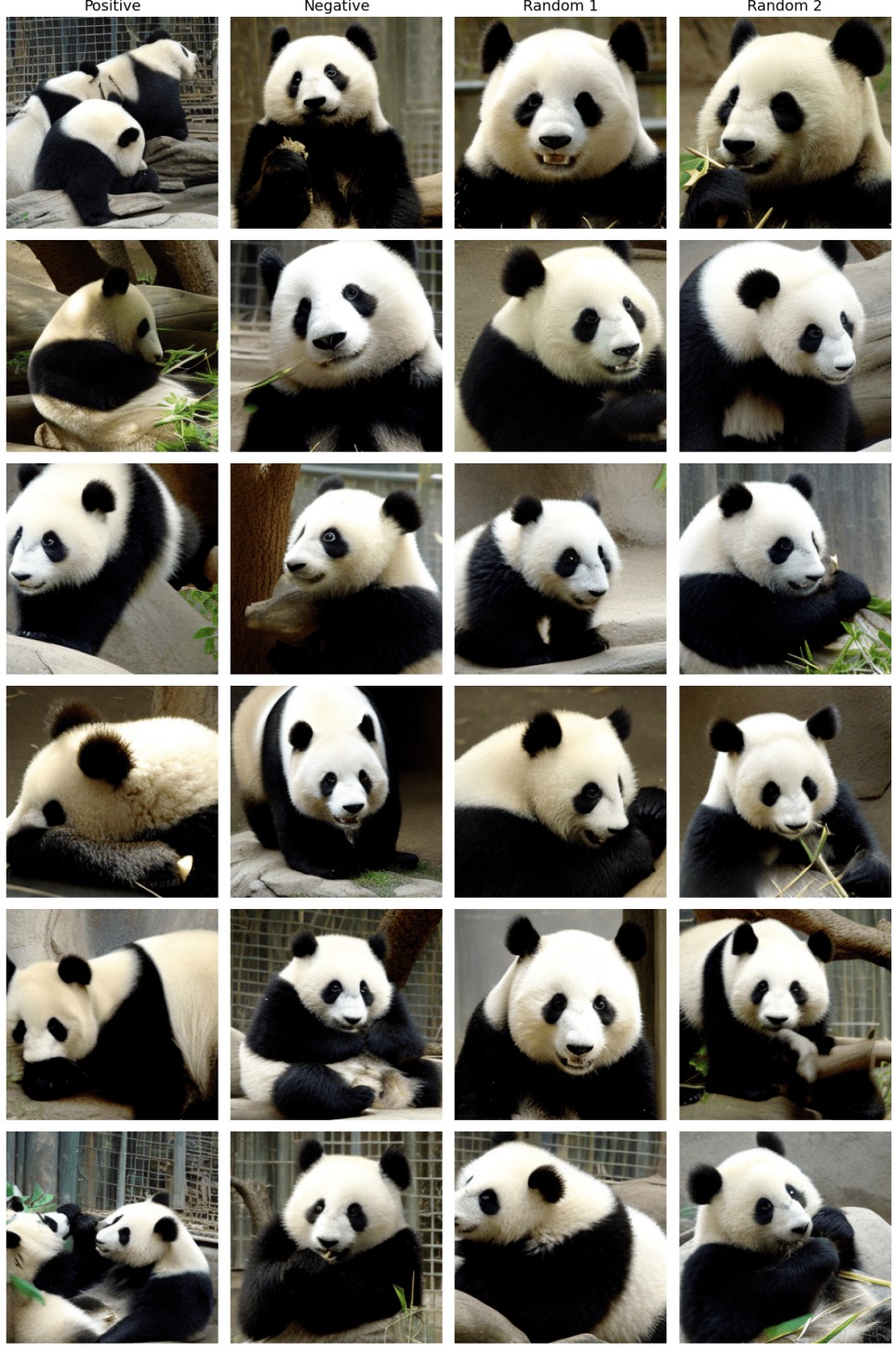

Figure 25: DiT Class 388: giant panda, panda

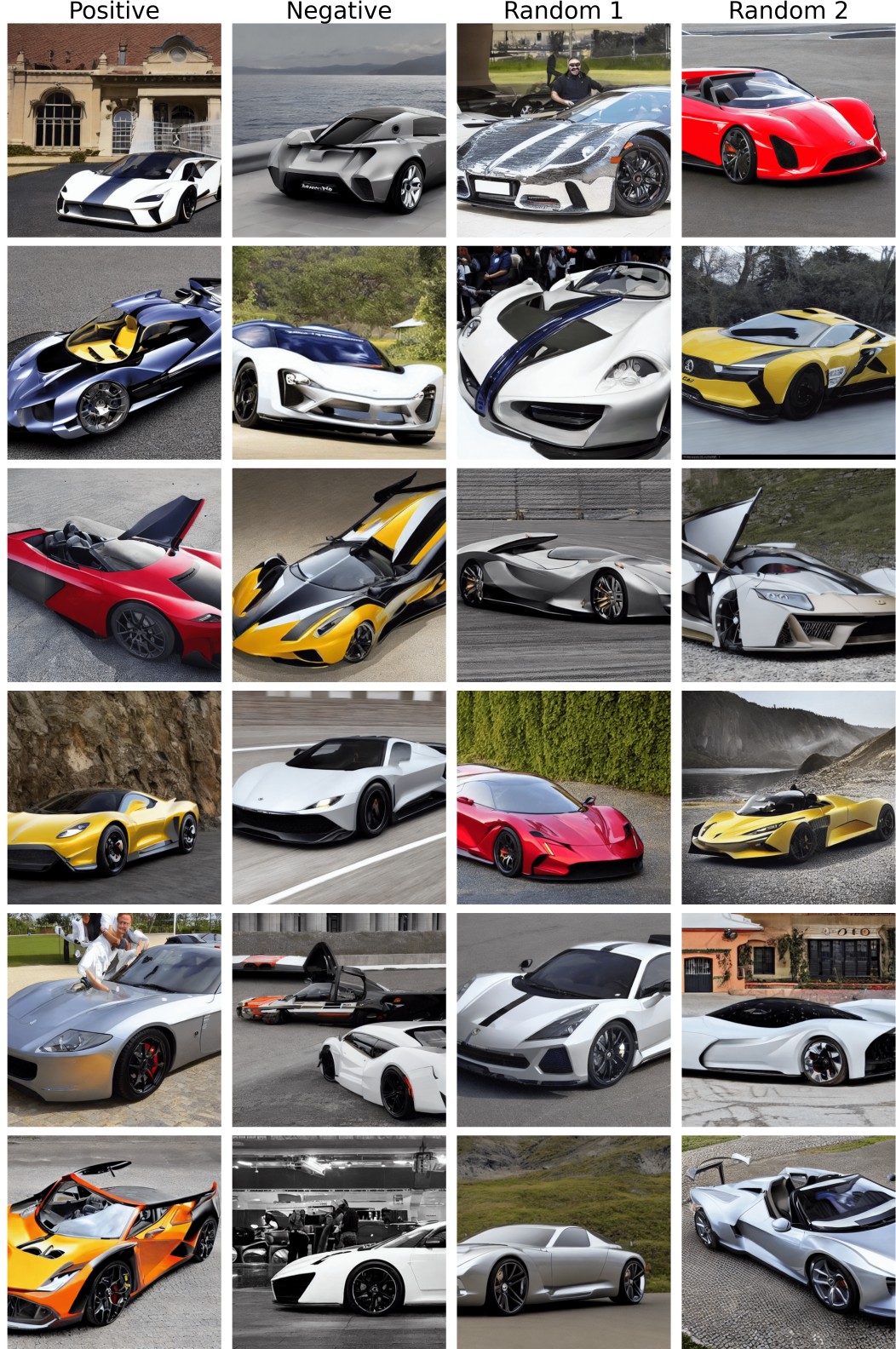

Figure 26: Prompt: "most expensive sports car"

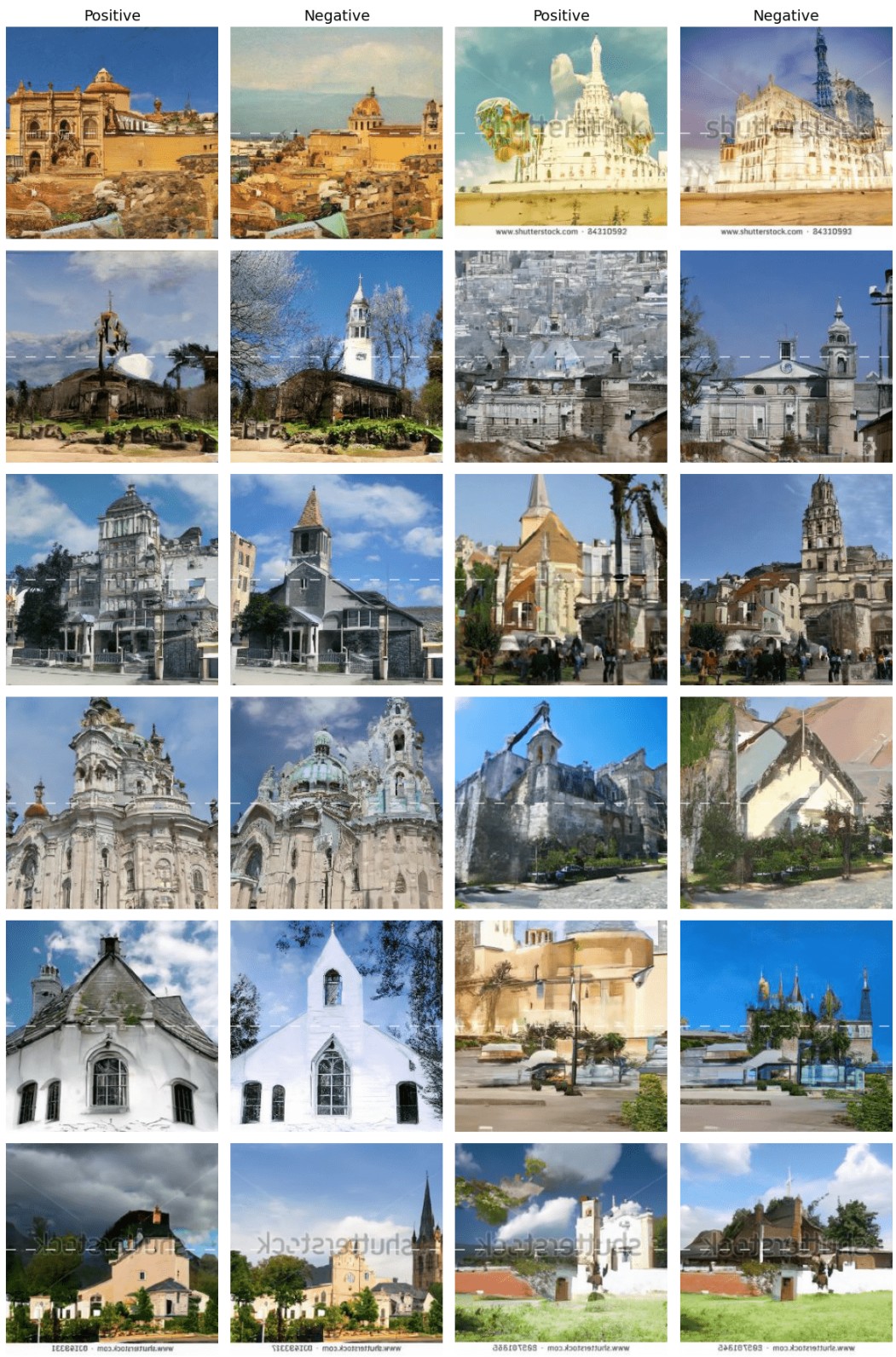

Figure 27: Partial Negation of LSUN-Church

Positive          Negative          Positive          Negative

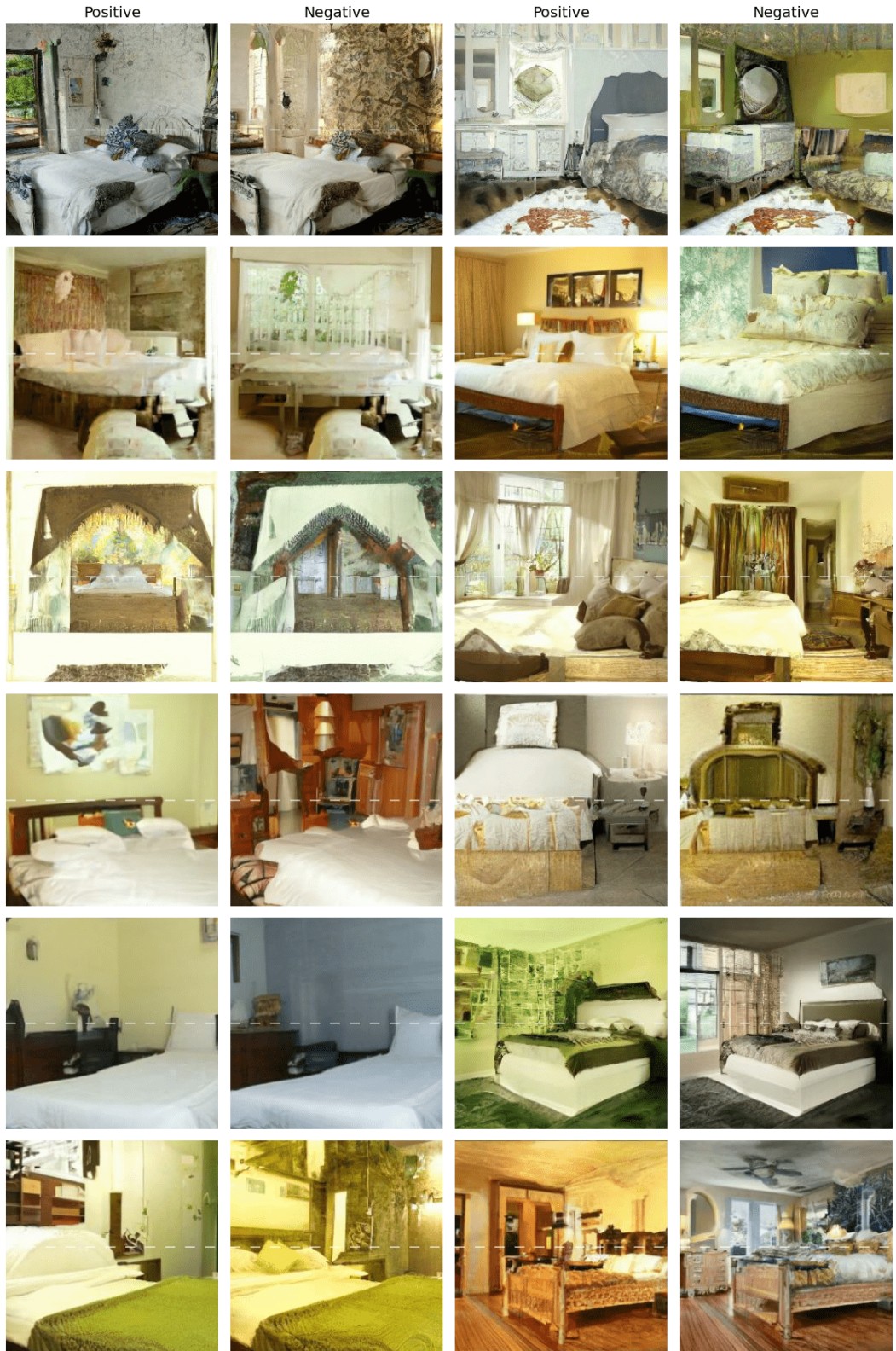

Figure 28: Partial Negation of LSUN-Bedroom

