# OpenReview forum: "Antithetic Noise in Diffusion Models"
_ICLR.cc/2026/Conference — ICLR 2026 Poster_

### Official Review · Reviewer_NGgz · 2025-10-24

**Soundness:** 4
**Presentation:** 4
**Contribution:** 3
**Rating:** 8
**Confidence:** 4

**Summary:**

The paper investigates the effect of sampling noise in generative image models by pairing each noise vector with its negation and analyzing the resulting generated images. The authors find that this antithetic noise approach produces samples that are strongly negatively correlated, revealing an approximate antisymmetry in the learned score function of diffusion models. They verify this symmetry statistically and propose it as a general property of generative models, extending beyond diffusion to VAEs and normalizing flows. Leveraging this finding, they show that antithetic noise can substantially improve uncertainty quantification, yielding tighter and more reliable confidence intervals.

**Strengths:**

1. The paper is thorough both empirically and theoretically, providing strong support for its claims through extensive experiments and proofs.

2. The results are substantial and appear to be novel, as the proposed method generalizes across different classes of generative models for image synthesis.

3. The proposed antithetic noise technique requires no retraining, making it practical and simple to apply.

4. The paper offers a compelling theoretical explanation for the observed symmetry through the antisymmetry of the score function, which deepens understanding of diffusion model behavior.

**Weaknesses:**

1. Some of the writing could be improved; for instance, the first sentence of the abstract (“We initiate…”) reads awkwardly.

2. The paper would benefit from an evaluation on OOD data, along with quantitative uncertainty metrics such as AUROC or AUPRC, to better demonstrate the method’s reliability in uncertainty estimation.

3. The work discusses uncertainty improvement but does not clearly distinguish between different types of uncertainty (e.g., aleatoric vs. epistemic), which would help clarify the scope and interpretation of the results.

4. Several relevant works on uncertainty in diffusion models are not cited:

[1] Berry, Lucas, Axel Brando, and David Meger. "Shedding light on large generative networks: Estimating epistemic uncertainty in diffusion models." The 40th Conference on Uncertainty in Artificial Intelligence. 2024.

[2] Berry, Lucas, et al. "Seeing the Unseen: How EMoE Unveils Bias in Text-to-Image Diffusion Models." arXiv preprint arXiv:2505.13273 (2025).

**Questions:**

1. Did you experiment with partial negation of the noise vector (e.g., negating only the first half or specific components of $z$) to analyze how localized symmetry affects the generated outputs?

2. Can you envision an analogous approach for LLMs, where an equivalent notion of antithetic sampling might reveal symmetry or uncertainty properties in text generation?

---

> ### Author Response · Authors · 2025-11-20
>
> Thank you for your positive evaluation and insightful feedback. We are encouraged by your recognition of the thoroughness of our empirical and theoretical work. Below, we address the questions, supported by additional experiments:
>
> **(1) Partial Negation of Noise Vectors**
>
> **In Appendix C.4.5,** We conducted experiments to examine how localized antithetic noise affects generated outputs. For each pair of generated samples, we used $Z_1$, drawn from a standard Gaussian distribution, and $Z_2$, obtained by duplicating $Z_1$ and negating only its upper half while leaving the lower half unchanged. We use unconditional diffusion models on LSUN-Church and LSUN-Bedroom, and generated 200 pairs for each dataset.
>
> The generated images exhibit spatial correspondence with the noise manipulation. As shown in the table, the top halves of each image pair show strong negative correlation (as expected), while the bottom halves remain highly positively correlated and visually nearly identical. Details are included in **Appendix C.4.5.**, visualizations are provided in **Figure 27-28**.
>
> | Dataset        | Top Half Standard Correlation | Bottom Half Standard Correlation | Top Half Centralized Correlation | Bottom Half Centralized Correlation |
> |----------------|-------------------------------|----------------------------------|----------------------------------|--------------------------------------|
> | LSUN-Church    | -0.40 (0.22)                  | 0.78 (0.11)                      | -0.58 (0.14)                     | 0.77 (0.11)                          |
> | LSUN-Bedroom   | -0.35 (0.27)                  | 0.82 (0.08)                      | -0.68 (0.14)                     | 0.79 (0.10)                          |
>
> This shows that the negative-correlation effect **acts locally** in noise space and directly affects the corresponding spatial regions in the generated images. Potentially, we can leverage this effect selectively in different parts of the generation space, which may enable new image-editing and diversity applications.
>
> Thank you for the insightful suggestion! We look forward to sharing further progress in future work.
>
> **(2) Extension to Language Diffusion Models**
>
> Excellent suggestion. We have also looked into recent diffusion-based language models. Language diffusion models employ diverse architectures, and our method could potentially be applied to those working with continuous embeddings, such as Diffusion-LM [1] and GENIE [2]. We plan to investigate if analogous symmetry properties exist in these settings.
>
> However, since text generation ultimately produces discrete tokens, the underlying mechanisms differ fundamentally from continuous image generation. Therefore, it is not straightforward to directly apply our antithetic noise approach to the generation process for discrete text sequences. We will explore potential adaptations that account for these architectural differences in our future work.
>
> **(3) Uncertainty Quantification: Scope and Clarification**
>
> Regarding the uncertainty quantification results, our method primarily targets **aleatoric uncertainty**, from the inherent randomness in the generative process due to noise sampling.
>
> By leveraging negatively correlated samples, we reduce the variance when estimating quantities derived from stochastic generation, such as pixel-level statistics (brightness, contrast) and PSNR, L1 reconstruction error.
>
> We select a confidence interval as a metric to capture the uncertainty in these estimates (e.g., "What is the expected PSNR?") rather than addressing epistemic uncertainty about the model's knowledge or OOD detection.
>
> This focus on aleatoric uncertainty may be particularly helpful in medical imaging applications, where clinicians need reliable uncertainty estimates for reconstructed MRI/CT scans to inform diagnostic decisions.
>
> **(4) Writing Improvements**
>
> We have revised the opening sentence of the abstract and made similar adjustments throughout the paper to improve clarity and flow.
>
> **(5) References**
>
> Thanks for pointing out these references. We have now added them to **Section 4** in the revised manuscript.
>
> Thank you again for the helpful feedback and suggestions.
>
> [1] Diffusion-lm improves controllable text generation. Li et. al.  NeurIPS 2022
>
> [2] Text generation with diffusion language models: A pre-training approach with continuous paragraph denoise. Lin et. al. ICML 2023

---

> > ### Comment · Reviewer_NGgz · 2025-11-25
> >
> > Thank you for your comments and for running the additional experiment to address the questions I raised. I maintain my score and fully support the paper's acceptance into ICLR 2026.

---

> > > ### Author Response · Authors · 2025-11-26
> > > **thanks!**
> > >
> > > Thanks for your thoughtful suggestions and support! We’re glad this discussion helped us see something new, and we’re happy to answer any further questions you may have.

---

### Official Review · Reviewer_mwy8 · 2025-10-30

**Soundness:** 3
**Presentation:** 3
**Contribution:** 1
**Rating:** 2
**Confidence:** 4

**Summary:**

This paper investigates the use of antithetic samples in the context of diffusion models. After presenting observational evidence in the form of pixel-wise correlation statistics of generated images across different datasets and models, the authors formulate the conjecture that the learned score network is an odd function up to a constant shift at every fixed time t.

Motivated by the negative correlation of samples generated from antithetic noise samples, the authors propose using antithetic sampling in diffusion models for various tasks such as uncertainty quantification and inverse problems.

Finally, they show that an antithetic Monte Carlo scheme significantly outperforms classical MC in terms of confidence interval width and overall sample diversity.

Overall, this paper reads as an advanced study of an expected statistical phenomenon underpinned by a well-known variance-reduction technique, rather than a fundamentally novel contribution.

**Strengths:**

- The paper is clearly presented and easy to follow.
- The underlying principle is simple and elegant and can be implemented easily.
- The conjecture is simple and relevant to the evidence presented.
- The experiments are thorough and convincing in terms of the superiority of antithetic vs independent sampling for uncertainty quantification and posterior sampling.

**Weaknesses:**

- The novelty feels quite limited. Antithetic sampling is a well-established variance-reduction technique, and the fact that fully negatively correlated samples passed into what is essentially a flow map yield back partially negatively correlated outputs feels like expected behaviour, not really a "universal discovery".

- The theoretical justification of the conjecture in the high-noise regime, although nicely presented, is fairly trivial since it mostly relies on the well-known fact that the score function of a Gaussian distribution is linear.

- The paper gives very little insight into what might be happening close to the target distribution p_0 (i.e., the data manifold). The omission feels significant, as it is clearly the region where getting the score function right matters most. Figure 3 clearly shows that the strong negative correlation breaks down faster as we approach the data distribution.

- Instead of focusing on the conjecture as a pure observational fact, the paper could benefit from either going deeper in the theoretical justification of the conjecture (in the likes of what is done using the FKG inequality in the appendix), or going deeper in the direction of using control variates to improve the efficiency of statistic computation/uncertainty quantification of diffusion models.

**Questions:**

- The benefits from the conjecture are not stated explicitly, e.g., how could one benefit from this underlying symmetry when designing/training a diffusion model?

- Are there ways to further validate the conjecture in the small-noise regime from a theoretical perspective?

- Could the authors propose hypotheses about the underlying mechanisms (either due to training algorithm/data/architecture) that might lead to the conjectured behaviour?

- Could other control variate schemes be explored?

---

> ### Author Response · Authors · 2025-11-20
> **1/2 major updates**
>
> Thank you for your constructive feedback!
>
> We are encouraged that you find our method clear, experiments convincing. The revised paper (uploaded as the new PDF) now directly addresses your three main concerns: (1) the theory is strengthened using Hermite polynomials and FKG inequalities; (2) a new proof is added to show that the forward process preserves all orthogonal symmetry; (3) the 2-point antithetic noise design is generalized to a $K$-point noise design with equal correlation $-1/(K-1)$ to provide further variance reduction.
>
> **1.  Novelty & Mechanism:** We would like to respectively clarify our novelty. *(1) Antithetic sampling is classical, but its behavior inside the diffusion model is not straightforward*. A diffusion process is a sequence of highly non-linear transformations. It is *not* obvious a priori that negative correlation in the noise space ($z, -z$) would survive this long and nonlinear process to generate negatively correlated images. *(2)The structure of the initial noise space in diffusion models has only recently begun to receive attention, and its role remains under-explored.* Our work provides what we believe is the first systematic study showing that this negative correlation survives throughout the entire denoising chain, and we offer both empirical and theoretical explanations for why this happens. *(3) On the application side, this negative correlation gives a practical tool for uncertainty quantification and posterior sampling in inverse problems.* We view this application-driven aspect as another novelty of the work.
>
> Nevertheless, we agree that a deeper theoretical explanation would significantly strengthen the paper. As suggested, to further explain this phenomenon, we added **Appendix B.6.1**: We prove that under an $MTP_2$ *(Multivariate Totally Positive of order 2)* condition and curvature bounds, DDIM preserves the negative correlation of antithetic inputs. This provides the "mechanistic" explanation: the sampler’s structure actively preserves the sign of the correlation despite the non-linearities.
>
> **2. Improving the Theory**
> We expanded the theory in two directions:
>
> • *Hermite Analysis:* **(App. B.3)** We now use Hermite polynomial expansion to prove quantitative, **monotone convergence** results of the density ratio and Fisher information toward the Gaussian limit. This is a quantitative bound showing that the score becomes progressively closer to a linear function.
>
>
> • *Symmetry Preservation:* **(App. B.4)** We proved a new theorem that helps explaining the $t\approx 0$ case: *Any orthogonal symmetry of* $p_0$ *is preserved by the forward OU process.*
>
>
> • *Relevance:* In the idealized setting where $p_{0}$ is reflection-symmetric, the density is even and the corresponding score is odd. Our new result shows that this symmetry is preserved for every $t$. In practice, real datasets are not globally symmetric. Nevertheless, the value of the ideal case is that many datasets contain small or local patterns that become nearly symmetric once early noise is added in the forward process. This initial noise rapidly smooths the distribution, and the remaining forward evolution drives $p_t$ toward a Gaussian at an exponential rate. As a result, the score $s_t$ moves toward a linear function in a controlled and monotone manner, which explains why an approximately odd score still emerges empirically even when exact symmetry is absent.
>
> 3. **K-antithetic noise design:**
>
> As suggested, in **Section 4 and 5**, we study a new control variate method. We can generalize the antithetic noise pair to a collection of $K$ noise variables, constructed so that every pair has the same negative correlation $-1/(K-1)$. This reduces variance further than pairwise sampling for uncertainty quantification tasks without retraining. The experiment uses 400 independent batches of $K=8$ negatively correlated noises (matching the 3200 image budget). The new AMC with $K=8$ achieves comparable or superior efficiency to our original AMC ($K=2$). For example, for the CelebA dataset and the brightness metric,  $K=8$ has a significantly higher efficiency of 130.69 vs  $K=2$’s 47.41. Details of the numerical results are included in **Section 5** of our new version.

---

> > ### Author Response · Authors · 2025-11-20
> > **2/2 response to questions**
> >
> > 4. **Response to Specific Questions**:
> >
> >      **Q1. Benefits of the symmetry**: We outline four possibilities:
> >
> >     - **Regularization.** One can add a small regularizer that encourages affine antisymmetry in the score. Since the empirical models already satisfy this structure to certain extent, regularizing it may improve stability or reduce variance in downstream tasks.
> >
> >     - **Mechanistic studies.** Structural constraints on the score may help improve recent mechanistic studies of diffusion models, which has recently attracted much attention [1-3]. Our empirical findings on score symmetry may provide a useful starting point for such analyses and could help guide future theoretical studies on generalization.
> >
> >     - **Hyperparameter tuning.** In practice, deploying a pre-trained diffusion model requires selecting multiple hyperparameters: step size, number of steps, CFG scale, etc. For instance, consider an image restoration algorithm $f_\phi$, where $\phi$ denotes the hyperparameter. For each $\phi$, we run the algorithm on a validation set and estimate its error $E(\phi)=\ \mathbb{E}\big[\|f_\phi(x) - x_{\mathrm{clean}}\|^2\big]$. This expectation is typically approximated by averaging over multiple random noises. When hundreds of candidate $\phi$’s must be evaluated, this Monte Carlo cost dominates the tuning process. Any method that achieves the same accuracy with fewer samples would therefore be quite valuable.
> >
> >     - **Variance reduction during training.** Training the score network $\epsilon_\theta^{(t)}$ by minimizing the score matching loss requires estimating an expectation over Gaussian noise, which is typically done using i.i.d. Monte Carlo samples. Our study suggests that antithetic sampling could help reducing the variance when computing the training loss, thus stabilizing the optimization.  Alternative forms of control variates such as the $K$-antithetic design and the ones discussed below could also be of interest.
> >
> >     **Q2. Further validate the conjecture in the small-noise regime from a theoretical perspective?**
> >
> >     -  This is an important open direction. We think a direction worth exploring is to combine (i) a structural study of $p_0$ and (ii) stability properties of the Ornstein–Uhlenbeck semigroup.  A common view is that data distributions are well-approximated by Gaussian mixtures. In many image datasets, these components often appear in approximately symmetric configurations (e.g., pairs of modes that are roughly mirror images). If one can formalize this as an “approximately symmetric” Gaussian mixture, then it becomes plausible to prove that the score also has approximate symmetry at $t = 0$. Our new result in App B.4 shows that *exact* orthogonal symmetries of $p_0$ are preserved for all $t$. A natural next step is to combine this with a perturbation analysis of the OU semigroup. In particular, one can treat a realistic $p_0$ described above as a small perturbation of an exactly symmetric mixture, and then study how the induced perturbation evolves. We expect that this would yield quantitative “stability” bounds showing that the score remains close to symmetric when the initial asymmetry is small.
> >
> >
> >
> >     **Q3. Hypothesis on mechanism**
> >
> >     - Great question.  Our current view is: (1) many real datasets have approximate reflection or permutation symmetry, and (2) even without explicit constraints, common architectures (e.g., U-Nets with skip connections) may produce locally linear behavior. We welcome further discussions
> >
> >     **Q4. Other control variates**
> >
> >    -  Apart from the $K$-antithetic design that is mentioned above, we also identify a broad class of symmetry-based control variates that may warrant further study.
> >
> >       The anti-symmetry of the score function motivated our proposal of antithetic sampling. Naturally, other symmetry properties in the score could suggest alternative control variates. As established in Appendix B.4, orthogonal symmetries in the data distribution are preserved during the forward noising process. For image distribution, one expects a certain degree of invariance under permutations or, more generally, rotations. Such rotational symmetries suggest generating $K$ Gaussian noise vectors by cyclically rotating one Gaussian vector, thereby forming an orbit under the symmetry group. The resulting $K$ images might lead to further variance reduction.
> >
> >
> > **Summary**
> >
> > Thank you again for the detailed suggestions. The revised paper now includes new theoretical results, extended experiments, and fuller discussion of the conjecture, specifically addressing the concerns you raised. We’d love to hear your thoughts.
> >
> > [1] A Good Score Does not Lead to A Good Generative Model, Li et. al., ArXiv 2024
> >
> > [2] An analytic theory of creativity in convolutional diffusion models, Kamp and Ganguli, ICML 2025
> >
> > [3] Towards a Mechanistic Explanation of Diffusion Model Generalization, Niedoba et al., ICML 2025

---

> ### Comment · Reviewer_mwy8 · 2025-11-25
> **Response to Authors' Rebuttal**
>
> Thank you for the detailed rebuttal and for the substantial revisions to the manuscript. I am satisfied with the addition of the more general $K$-antithetic sampling scheme, which strengthens the practical contribution of the work. The expanded theoretical analysis also provides additional support for the conjecture in the high-noise regime. For these reasons, I have increased my score.
>
> That said, I still have reservations regarding the novelty and, in particular, the theoretical grounding of the conjectured symmetry. While I understand that, from a black-box perspective, there is no guarantee that a long sequence of neural-network–based transformations should preserve negative correlation, diffusion samplers do not behave like arbitrary non-linear maps. Given their structure and the objective of learning score functions associated with a specific diffusion process, it seems reasonable to expect some degree of correlation preservation, and thus, this phenomenon does not strike me as especially surprising.
>
> In addition, although Appendix B.4 is a nice step toward understanding the small-noise regime, it ultimately offers only limited evidence for the conjecture where the score network is most non-linear. The plan you outline in your response to Q2, that is to say, combining structural assumptions on $p_0$ with stability properties of the OU semigroup, would have provided a very compelling theoretical justification had it been developed further in the paper.
>
> Overall, I appreciate the improvements in the revised submission, but these points still temper my assessment of the conceptual contribution.

---

> ### Author Response · Authors · 2025-11-26
> **thanks and clarification**
>
> Thank you for your follow-up and for increasing your score. We appreciate the careful reading and agree that understanding the small-noise regime is an important next step. We plan to develop the structural–OU stability direction outlined in the rebuttal in a separate follow-up work (and appreciate the inspiration!). Your feedback has helped us refine the scope and future direction of this line of research.
>
> We emphasize the novelty of this work is the first discovery of this universal phenomenon, an explanation of why it appears, and its practical use.

---

### Official Review · Reviewer_JEpj · 2025-10-31

**Soundness:** 3
**Presentation:** 3
**Contribution:** 3
**Rating:** 6
**Confidence:** 3

**Summary:**

This paper shows that in the diffusion model sampling, pairing each noise sample with its negation induces a strong negative correlation in the generated outputs.  The authors also note that this effect is universal, observed across datasets and architectures, and even in other generative models such as VAEs and normalizing flows. They claim both empirical and theoretical support for a symmetry conjecture that the learned score function exhibits approximate affine antisymmetry. The resulting negative correlation enables substantially improved uncertainty quantification, significantly reduces the uncertainty, suggested by  90% narrower confidence intervals

**Strengths:**

1. The study of antithetical initial noise in diffusion models appears new to me.
2. Although the empirical results are not perfectly aligned with the theoretical argument in Lemma 1, the characterization is interesting and could inspire future studies.
3. The paper is well-written and easy to follow.

**Weaknesses:**

1. The work would be more complete if the authors could provide some discussion on why the score function admits this affine antisymmetric property.
2. While the work shows that the found fact could result in new methods to do uncertain quantification with significantly narrower CI, as we deal with the generative models, we can potentially generate an infinite number of samples, which potentially drops the significance of the proposed methods.
3. I am a bit confused about the results in Table 1. It appears that models and datasets are mixed up.

**Questions:**

Please refer to the weaknesses.

---

> ### Author Response · Authors · 2025-11-20
>
> Thank you for your positive review! We are happy that you found our method novel, the idea inspiring, and the writing easy to follow. We address the questions below.
>
> **1. More explanation on the symmetry:**
>
> Great point. Following your suggestion, we have added new theoretical results in Appendices **B.3, B.4, and B.6.1.**
> In particular, we added two new components:
>
> (1.a) a quantitative Hermite polynomial-based analysis of the OU flow showing monotone convergence of the density ratio and Fisher information,
>
> (1.b) a symmetry-preservation result that explains when odd-score structure should persist for all $t$.
>
> For the symmetry effect, the combination of (1.a) and (1.b) provides a coherent explanation:  In the idealized setting where $p_{0}$ is reflection-symmetric, the density is even and the corresponding score is odd. Our new result shows that this symmetry is preserved for every $t$. In practice, real datasets are not globally symmetric. Nevertheless, the value of the ideal case is that many datasets contain small or local patterns that become nearly symmetric once early noise is added in the forward process. This initial noise rapidly smooths the distribution, and the remaining forward evolution drives $p_t$ toward a Gaussian at an exponential rate. As a result, the score $s_t$ moves toward a linear function in a controlled and monotone manner, which explains why an approximately odd score still emerges empirically even when exact symmetry is absent.
>
> ---
>
> **2. More samples always lead to accurate estimates:**
>
> Good point, thank you for the comment! We fully agree that, by the law of large numbers, an arbitrarily large number of samples will converge to the true expectation, so all methods considered here are *consistent*.
>
>
> Nevertheless, generating these samples has a real cost. Each call to a modern diffusion model is expensive, and this cost grows quickly for high-resolution or high-dimensional data such as 3D images, microscopy data, or video, where sampling is far more demanding. Many scientific computing tasks also fall into this setting and must operate under strict compute limits. Reducing the number of required samples therefore brings a direct reduction in runtime and resource use. In our experiments, the method gives large gains: for linear statistics it often cuts the compute cost by a factor of 10 to 100, and for diffusion inverse solvers the savings range from about 34% to 84%. We hope these gains provide a practical and meaningful reduction in compute cost and resource use across these settings.
>
> We provide two potential use cases:
>
> 1. First, suppose we are tuning a diffusion-based image restoration algorithm $f$ with many hyperparameters: step size, number of reverse steps, guidance scale, noise schedule, etc. For each configuration $\theta$, we run the algorithm on a validation set and estimate its mean squared error $E(θ)=\; \mathbb{E}\big[\|f_\theta(x) - x_{\mathrm{clean}}\|^2\big],$ where the expectation is over the diffusion initialization. In practice, you approximate $E(\theta)$ by averaging over $k$ initializations. When there are dozens or hundreds of candidate $θ$’s, this Monte Carlo cost dominates the tuning process, so any method that gives the same accuracy with fewer Monte Carlo samples is extremely valuable.
> 2. Another example comes from benchmarking diffusion-based inverse solvers. For a fixed diffusion prior, there are many solver designs, and testing all of them can be costly. Evaluating any solver requires estimating the expected value of an error metric, such as the L1 error or PSNR. This makes our method a more efficient tool for comparing and benchmarking the full set of existing solvers.
>
> ---
>
> **3. Table 1**:
>
> Great point. We have updated Table 1 to include separate columns for the model and the dataset. Thanks!

---

### Official Review · Reviewer_cieF · 2025-11-01

**Soundness:** 3
**Presentation:** 3
**Contribution:** 3
**Rating:** 6
**Confidence:** 2

**Summary:**

The paper discovers a universal negative correlation in outputs from diffusion models when pairing each initial Gaussian noise  with its negation (“antithetic noise”). This holds across datasets, architectures (U-Net, DiT), unconditional/conditional sampling, DDIM / DDPM, and even VAEs / Normalizing Flows. The authors explain the phenomenon with a symmetry conjecture: the learned score function is approximately affine antisymmetric (odd up to a constant), supported by temporal correlation analyses, 1D slices of score outputs, and theory in the high-noise regime via the OU process. Leveraging the strong negative correlation, they propose antithetic Monte Carlo for uncertainty quantification, yielding up to 90% tighter confidence intervals and large efficiency gains, and show complementary benefits from randomized QMC. Applications include estimating pixel-wise statistics, evaluating diffusion inverse solvers (DPS/DDS), and improving diversity, as well as training-free and model-agnostic editing.

**Strengths:**

(i) The paper proposes a simple, training-free, model-agnostic procedure with no runtime overhead

(ii) The paper features broad empirical validation across datasets, samplers (DDIM/DDPM), architectures, and even VAEs/flows

(iii) Antithetic Monte Carlo is well motivated and yields substantial, measurable variance reduction

(iv) Beyond the main paper, the work provides good reproducibility details and extensive appendices

**Weaknesses:**

(i) While theoretical support for the symmetry conjecture is partial and focused on high-noise regimes, the symmetry conjecture remains unproven in generality. The novelty lies mostly in documenting the phenomenon antithetic variance reduction and its strength in diffusion models.

(ii) Many results use pixel-level correlations and simple statistics, while semantic-level uncertainty and quality metrics (e.g., FID, CLIP alignment) are less explored or could be expanded upon more.

(iii) Conditional settings (e.g., strong CFG guidance, complex prompts) and additional methods (e.g., Consistency Models) are not exhaustively studied.

**Questions:**

In addition to the weaknesses outlined in points (i-iii), I present the following questions for the authors to address:

(1) How does the negative correlation behave under varying classifier-free guidance scales and different guidance strategies?

(2) Can the method improve semantics in text-image alignment uncertainty, and object counts/ positions beyond pixel statistics (a qualitative study would suffice)?

(3) How does antithetic pairing interact with distillation methods (e.g., Consistency Models) and very few-step generation (few-step generation would suffice if not applicable to consistency models)?

(4) While results seem promising, I miss a detailed discussion of limitations. Are there scenarios where antithetic pairing reduces diversity or text adherence?

---

> ### Author Response · Authors · 2025-11-20
>
> Thank you for your thoughtful review and positive feedback. We address the concerns and questions below, supported by additional theory and experiments:
>
> **(1) Negative Correlation Under Varying CFG Scales**
>
> In **Appendix C.4.4**, we did experiments on both SD1.5 and DiT across CFG scales {1, 3, 5, 7, 9}. For each setting, we generated 100 Positive and Negative (PN) and Random and Random (RR) pairs for 25 prompts/classes.  For all cases, negated noise continues to produce strongly negatively correlated samples. Meanwhile, the raw correlation for both PN and RR grows as CFG increases. This can be explained as follows: larger CFG values pull samples more strongly toward the conditioning signal (prompt/class), thus shrinking the space of plausible outputs. As samples concentrate more tightly, they become more similar to one another. Thus, the correlations of both PN and RR pairs increase with CFG, while PN remains more negative than RR.
>
> **(2) Consistency Models**
>
> In **Section 3.1**, we tested unconditional Consistency Models (CAT and Bedroom) using 1,600 PN/RR pairs for each dataset. Both models exhibit strong and statistically significant negative correlation in PN pairs: the Cat model shows a centralized mean correlation of –0.91 (t ≈ –224), and the Bedroom model shows –0.84 (t ≈ –190), while RR pairs remain centered near zero in both cases. These results show that the antithetic structure is preserved through distillation.
>
> Interestingly, the correlations for consistency models appear more negative than for diffusion models, which might be a promising future direction.
>
> **(3) Will it reduce metrics such as diversity and text adherence?**
>
> Good question. For any diffusion model DM and any **single-image** metric  $S : \text{Image} → \text{R}$ (e.g., MUSIQ or CLIP with respect to a fixed prompt), we can prove that  $\text{DM}(z)$ and $\text{DM}(-z)$ have the same distribution, and thus the same expectation. Hence, negating the noise does not change the expected value of such metrics. As an additional experiment, we evaluated SD1.5-generated images using CLIP scores and MUSIQ. The table below reports the average scores:
>
> | Metric | Pos | Neg | Rand1 | Rand2 |
> | --- | --- | --- | --- | --- |
> | CLIP | 30.19 | 30.20 | 30.18 | 30.18 |
> | MUSIQ | 70.19 | 70.14 | 70.14 | 70.13 |
>
> Both PN pairs and RR pairs produce indistinguishable CLIP and MUSIQ scores, indicating that antithetic pairing does not degrade semantic quality or text–image alignment.
>
> Regarding **population-level metrics** (such as diversity), the behavior can be more subtle. In our current experiments, we do not observe any reduction in global sample diversity or prompt adherence.
>
> **(4) Beyond pixel statistics:**
>
> Good question. We ran additional experiments on new metrics, e.g. MUSIQ on CIFAR-10 and CLIP-based text alignment on SD 1.5 (100 prompts). The relative efficiency of the AMC estimator compared to MC is about 1.05 for MUSIQ and about 1.025 for CLIP.  These results show that our method still yields a reduction in uncertainty for these metrics, but the gains are more modest than those observed for pixel-based statistics. We also discuss this in (6).
>
> **(5) Additional theoretical support:**
>
> We have added new theoretical results in Appendices **B.3, B.4, and B.6.1.**
> For **symmetry beyond large t**: we added two new components:
>
> (1.a) a quantitative Hermite polynomial-based analysis of the OU flow showing monotone convergence of the density ratio and Fisher information,
>
> (1.b) a symmetry-preservation result that explains when odd-score structure should persist for all $t$.
>
> For the **symmetry effect**, the combination of (1.a) and (1.b) provides a coherent explanation: In the idealized setting where $p_{0}$ is reflection-symmetric, the density is even and the corresponding score is odd. Our new result shows that this symmetry is preserved for every $t$. In practice, real datasets are not globally symmetric. Nevertheless, the value of the ideal case is that many datasets contain small or local patterns that become nearly symmetric once early noise is added in the forward process. This initial noise rapidly smooths the distribution, and the remaining forward evolution drives $p_t$ toward a Gaussian at an exponential rate. As a result, the score $s_t$ moves toward a linear function in a controlled and monotone manner, which explains why an approximately odd score still emerges empirically even when exact symmetry is absent.
>
> For **negative correlation**, we added **Appendix B.6.1**: We prove that under an $MTP_2$ (Multivariate Totally Positive of order 2) condition and curvature bounds, DDIM preserves the negative correlation of antithetic inputs.
>
> **(6) Limitation discussion:**
>
> Great suggestion. We have now expanded Section 6 to discuss limitations, including a discussion on non-pixel statistics.
>
> We hope these new theoretical results, expanded experiments, and additional discussion help clarify the core phenomenon and address your concerns.

---

### Comment · Area_Chair_Latk · 2025-11-21

Dear Reviewers,

We kindly encourage you to review and respond to the authors’ rebuttals. Your timely feedback is important for ensuring a fair and thorough review process. Thank you for your contributions to ICLR 2026.

AC

---

### Author Response · Authors · 2025-12-03
**Summary of Reviews and Post-Rebuttal Revisions**

Dear AC,

Thank you for handling our submission. Below we summarize the reviews and how we have addressed the points they raised.

**Part 1: Before rebuttal: Summary of strengths and concerns.**

Our submission initially received scores of 8, 6, 6, and 2. All four reviewers found the paper clear and the empirical study thorough. Reviewers *NGgz* (score 8) and *cieF* (score 6) highlighted the broad empirical evidence across {datasets, architectures, and samplers}, the training-free nature of the method, and the strong practical gains in uncertainty quantification. Reviewer *JEpg* (score 6) highlights the novelty of the work and notes that the theoretical results may inspire further research. Reviewer *mwy8* (initial score 2) finds the underlying principle elegant and the method practical. Reviewers also noted that the theoretical components are well presented and that the phenomenon is consistent and stable across models.

The main questions and concerns were:

(i) from *mwy8* and *JEpj:* more **discussion of the symmetry mechanism**, especially in the small-noise regime;

(ii) from *cieF* and *mwy8:* deeper **theoretical support**;

(iii) **additional experiments** on (1) distilled models and (2) the effect of classifier-free guidance scale (from *cieF*), and (3) partial negation of the noise (from *NGgz*);

(iv) from *mwy8*, alternative **variance reduction methods;**

(v) from *mwy8*, questions about the level of novelty and **whether the negative-correlation effect should be viewed as expected**.

**Part 2: Our revisions & How we addressed reviewers’ concerns.**

Questions (i) and (ii) both relate to the theoretical foundation of the phenomenon. In response, we added theory based on Hermite expansions and monotone convergence of density ratios and Fisher information, and proved a symmetry-preservation result showing that any orthogonal symmetry in the data distribution is maintained by the forward OU flow (App. B.3–B.4). We also show that negative correlation is preserved under DDIM (App. B.6.1), which gives a new mechanism-level explanation for the emergence of negative correlation.

To address Question (iii), we ran new experiments requested by reviewers, including varying classifier-free guidance scales, tests on Consistency Models, and partial-negation experiments showing spatially localized effects (Apps. C.4.4–C.4.5).

To address Question (iv), we extended the two-point antithetic noise setup to a $K$-antithetic design and added new experiments demonstrating that this broader scheme gives additional gains for uncertainty quantification.

To address Question (v), we emphasized in our rebuttal that although antithetic sampling is classical, it is not clear a priori (1) that negative correlation in the noise space would persist through the long sequence of highly non-linear diffusion updates, and (2) that the extent of negative correlation would be so significant. Our novelty lies in first identifying this universal phenomenon, explaining why it occurs, and demonstrating its practical usefulness.

**Part 3: Status before the rollback**.

Nov 25 (2 days before the freeze), Reviewer *mwy8* (initial score 2) acknowledged that the generalized $K$-antithetic design “strengthens the practical contribution” and the strengthened theory “provides additional support”, and **increased their score** to reflect this. Their comments explicitly state this change.

Nov 25 (2 days before the freeze), Reviewer *NGgz* (score 8) maintained score and wrote that they **fully support** acceptance.

While the remaining two reviewers (both with initial score 6) did not have the chance to participate in the post-rebuttal discussion, we believe our revisions address all their stated concerns. **For reviewer *cieF***, our new experiments on consistency models, varying CFG scale, and non-pixel statistics directly address their questions about distilled models, CFG, and semantic-level uncertainty, respectively. The added theoretical results directly respond to their question for additional theoretical support. **For reviewer *JEpj***, our added App. B.3-4, App. B.6.1 directly answers the question on more discussion of symmetry. Our rebuttal directly addressed the question on the benefits of narrower confidence interval by clarifying that our method yields significant compute savings in a variety of downstream applications.

**Part 4: Concluding remarks.**

This work identifies a universal, reproducible, and significant negative-correlation phenomenon across diffusion models, provides theoretical explanations for why it appears, and turns it into a practical tool that yields large efficiency gains with no retraining and no additional cost. Guided by the reviewers’ suggestions, the revision further strengthens both the theoretical and empirical contributions, and we believe the concerns raised in the reviews have been substantively addressed. We hope this summary is helpful for the AC’s recommendation.

---

### Meta-Review · Area_Chair_WyxL · 2026-01-08

**Summary:**

The paper presents a systematic study of antithetic initial noise in diffusion models, a simple,
training-free technique where each noise sample $z$ is paired with its negation.
The authors identify a phenomenon: the pairing consistently produces
negatively correlated samples across various datasets and  architectures and generative frameworks including VAEs and Normalizing Flow.
To explain this, the authors propose a symmetry conjecture, suggesting that learned score functions are approximately affine antisymmetric.
The authors exploits this finding to propose uncertainty quantification.

Most Reviewers have found that the paper have  strong merits.  They mostly praised its utility and efficiency as well
as the empirical validation. One reviewer had some concerns about the novelty  that have been cleared during rebuttal.
As such, reviewers have mostly positive views about the paper warranting acceptance.

**Reviewer Concerns:**

cleared:
- novelty

**Reviewer Scores:**

I am not able to answer this question

---

### Decision · Program_Chairs · 2026-01-26

Accept (Poster)